# DEMIX: Dual-Encoder Latent Masking Framework for Mixed Noise Reduction in Ultrasound Imaging

## Abstract

Ultrasound imaging is widely used in noninvasive medical diagnostics due to its efficiency, portability, and avoidance of ionizing radiation. However, its utility is limited by the quality of the signal. Signal-dependent speckle noise, signal-independent sensor noise, and non-uniform spatial blurring caused by the transducer and modeled by the point spread function (PSF) degrade the image quality. These degradations challenge conventional image restoration methods, which assume simplified noise models, and highlight the need for specialized algorithms capable of effectively reducing the degradations while preserving fine structural details. We propose DEMIX, a novel dual-encoder denoising framework with a masked gated fusion mechanism, for denoising ultrasound images degraded by mixed noise and further degraded by PSF-induced distortions. DEMIX is inspired by diffusion models and is characterized by a forward process and a deterministic reverse process. DEMIX adaptively assesses the different noise components, disentangles them in the latent space, and suppresses these components while compensating for PSF degradations. Extensive experiments on two ultrasound datasets, along with a downstream segmentation task, demonstrate that DEMIX consistently outperforms state-of-the-art baselines, achieving superior noise suppression and preserving structural details. The code will be made publicly available.

## 1 Introduction

Digital images are often degraded by noise arising from acquisition and transmission (Liu et al., 2017; Huang et al., 2017; Goyal et al., 2018). Noise can be attributed to various factors, including random thermal fluctuations of imaging sensors, inherent sensor imperfections, and interactions with the imaging environment, among others. The noisy images thus acquired have random intensity variations and degraded characteristics, which make the performance of automated downstream tasks such as segmentation and classification significantly challenging. Mathematically, image denoising can be defined as an inverse problem that aims to recover the true image from a noisy observation while preserving its structural details. Image denoising has been extensively studied in the past few decades (Buades et al., 2005; Kervrann & Boulanger, 2006; Ashouri & Eslahchi, 2022; Xie et al., 2024). Noise can be signal-dependent or signal-independent,

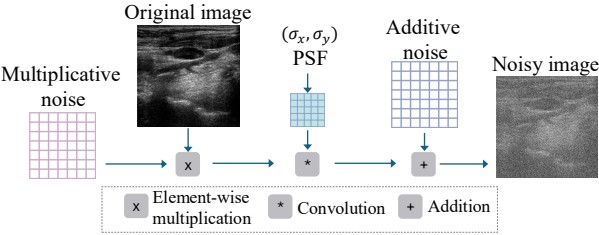

Figure 1: Noisy image generation: The original image is first multiplied with signal-dependent speckle, convolved with the PSF of the imaging system and finally added to signal-independent sensor noise.

depending on its source and the characteristics of the imaging device. The dominance of signal-independent

noise in natural illumination conditions motivates the standard additive Gaussian noise model for image denoising applications (Luisier et al., 2010). Although the simplified assumption leads to simpler analysis and algorithmic design, the assumption does not apply to many image sources. Images acquired from a wide class of imaging modalities are often degraded by various forms of noise, making the image denoising problem particularly challenging. To name a few, images acquired from ultrasound applications, fluorescence microscopy, and astronomy are affected by a mixture of different kinds of noise, which deviate significantly from the additive Gaussian noise assumption. The complex noise patterns arising in these images require specially designed denoising algorithms that go beyond simplistic models. Additionally, these images are further distorted by the point spread function (PSF) of the imaging system. The PSF characterizes the system's response to a point source and encapsulates effects such as diffraction, optical aberrations, and sensor imperfections. The final acquired image is a spatially distorted representation of the true underlying structure. Accurate PSF modeling has been a key research focus for image reconstruction and analysis, specifically from astronomical image analysis (Liaudat et al., 2023; Beltramo-Martin et al., 2020; Jia et al., 2020) to microscopy image analysis (Li et al., 2018; Stallinga et al., 2009; Shechtman et al., 2014; Zhang et al., 2007). In ultrasound images, the PSF has been widely utilized in a range of applications, including motion estimation in tissues (Saris et al., 2018), analytical methods for PSF estimation, deconvolution, and restoration algorithms (Dalitz et al., 2015; Yang & Ren, 2010; Zhao et al., 2016), PSF-informed super-resolution algorithms (Ploquin et al., 2015) and simulation frameworks with spatially varying PSFs (Varray et al., 2016; Schretter et al., 2018).

Ultrasound is a highly efficient, radiation-free, and non-invasive imaging modality for imaging soft tissues in diagnostic settings. Despite these advantages, ultrasound images are degraded by signal-dependent speckle noise, arising from the coherent interference of ultrasonic waves scattered by sub-resolution tissue structures. Speckle is an inherent characteristic of all coherent imaging systems, where the constructive and destructive interference of waves gives rise to speckle. Thus, speckle is seen in applications such as microscopy, astronomy, and sonar, among several other applications, and ultrasound is used as a representative example. The speckle, combined with the additive Gaussian noise caused by thermal fluctuations in acquisition sensors and the PSF-induced distortions, severely degrades the image quality and limits the performance of automated downstream algorithms. Addressing these challenges requires a unified framework to simultaneously mitigate additive and multiplicative noise components while reducing PSF-induced distortions.

**Contributions:** In this work, we propose DEMIX, the first PSF-aware dual-encoder diffusion-inspired denoising framework to simultaneously reduce additive and multiplicative noise from ultrasound images. The key contributions are:

- DEMIX leverages a novel dual-encoder strategy to separately model signal-dependent speckle and signal-independent Gaussian noise, leading to superior denoising performance.

- A latent masking mechanism combines the encodings of the two encoders with a masked, attention-like gating function to disentangle complex heterogeneous noise in the latent space while preserving structural details in the reconstructed images.

- Lateral and axial PSF distortions are encoded within each encoder, allowing the model to adaptively guide the denoising model to focus on the distorted regions.

- Extensive experiments on two ultrasound image datasets, including downstream evaluations, demonstrate the strong generalizability of DEMIX in handling varying noise combinations and PSF distortions. These results highlight the robustness and generalizability of DEMIX in diverse real-world ultrasound imaging conditions.

## 2 Related Works

**Image Denoising:** Image denoising has been extensively studied for different imaging modalities with classical optimization-based methods as well as modern deep learning architectures. Spatial domain filtering algorithms have been widely studied in (Wiener, 1964; Tomasi & Manduchi, 1998; Yang et al., 1996; Bouboulis et al., 2010; Benesty et al., 2010; Yang et al., 1995) and can be broadly categorized as linear

spatial filters (Gonzalez, 2009; Benesty et al., 2010) and non-linear ones (Yang et al., 1995; Tomasi & Manduchi, 1998). Traditional image restoration algorithms include total variation regularization (Rudin et al., 1992), non-local means (NLMeans) algorithm (Buades et al., 2011) and transform domain-based algorithms (Strang, 1999; Ahmed et al., 2006; Ergen, 2012). In the past decade, a significant number of deep learning algorithms have been proposed for image denoising applications (Izadi et al., 2023; Tian et al., 2020a). Convolutional neural networks (CNNs) have been used to denoise imaging from a wide range of image sources (Tian et al., 2020b; Lefkimmiatis, 2018; Zhang et al., 2021; Jin et al., 2019b). These algorithms include supervised, semi-supervised, and unsupervised methods that have been widely adopted to design cutting-edge, efficient algorithms (Zhou et al., 2020; Kim & Ye, 2021; Laine et al., 2019; Pang et al., 2021; Cui et al., 2019). Adversarial training has also been very dominant in the design of novel image denoising algorithms (Zhong et al., 2020; Chen et al., 2020; Lin et al., 2023; Khan et al., 2021). However, all these algorithms have been proposed for images that are corrupted with a single noise form, which in most cases is additive white Gaussian noise (AWGN). However, images obtained from many sensors are corrupted with complex noise patterns arising from a mixture of noise types.

**Mixed Noise Removal:** Real-world images are often degraded by complex, heterogeneous noise distributions arising from different aspects of the imaging device. Image restoration algorithms for hyperspectral images corrupted with mixed noise have been widely studied (Zhuang & Ng, 2021; Zhang et al., 2019a; Zheng et al., 2019; Luo et al., 2021; Wang et al., 2020; Rasti et al., 2019; Zhuang et al., 2021). Hyperspectral images are corrupted by signal-dependent Poisson noise, signal-independent additive Gaussian noise, and impulse noise (Rasti et al., 2018). Various approaches have been proposed for restoring hyperspectral images, such as subspace learning methods (Shi et al., 2024; Chen et al., 2024; Sun et al., 2018; Wu & Li, 2024; Cao et al., 2019; Wang et al., 2023; Dong et al., 2019), Gaussian mixture models exploiting spatial and spectral information (Zhuang & Ng, 2021; Houdard et al., 2018; Jin et al., 2019a) and recent neural network-based image denoising algorithms (Zeng et al., 2024; Dong et al., 2019). Beyond hyperspectral imaging, image enhancement algorithms have been widely studied for images corrupted with additive white Gaussian noise (AWGN) and impulse noise (Huang et al., 2017; Kim et al., 2020; Jiang et al., 2014; Liu et al., 2017), images corrupted by a mixture of Poisson and Gaussian noise (Luisier et al., 2010; Le Montagner et al., 2014; Mannam et al., 2022; Zhang et al., 2019b; Khademi et al., 2021). These methods include optimization methods, statistical modeling, and deep learning frameworks.

**Diffusion Models:** Diffusion models (Ho et al., 2020; Nichol & Dhariwal, 2021; Song et al., 2020) have recently emerged as the state-of-the-art models for image generation applications such as image inpainting (Lugmayr et al., 2022; Yang et al., 2023), image super-resolution (Sahak et al., 2023; Wu et al., 2023) and inverse problems (Chung et al., 2022; Daras et al., 2024). Diffusion models comprise a forward process and a reverse process, and the model learns to generate realistic data from an isotropic Gaussian, thus being able to model complex data distributions. However, most diffusion models rely on a single noise model, which significantly restricts their applicability to real-world data that are degraded by heterogeneous noise.

**Limitations of previous work:** Images acquired from a wide class of imaging modalities, such as ultrasound, synthetic aperture radar (SAR), and optical coherence tomography (OCT), are corrupted by a combination of signal-dependent speckle and signal-independent Gaussian noise. However, very few studies have considered this specific combination of heterogeneous noise corruption that is observed in coherent imaging systems. Conventional denoising algorithms, which are designed exclusively for additive or multiplicative noise, struggle to enhance these images. Designing a unified generalizable framework for signal-dependent speckle and signal-independent Gaussian noise remains an underexplored and challenging research area.

## 3 Background: PSF for Ultrasound Images

The point spread function (PSF) of the ultrasound imaging system can be assumed to be the impulse response of the system. The PSF $p(i, j)$ is linear and assumed here to be space invariant, where $i$ and $j$ represent the lateral and axial directions of the ultrasonic beam. The PSF $p(i, j)$ can be expressed as the

convolution of the one-dimensional functions $p_1(i)$ and $p_2(j)$, which represent the distortions along the lateral and axial directions, respectively. The lateral distortion function $p_1(i)$ accounts for the spatial response of the transducer aperture, which acts first as the transmitter and then as the receiver. The axial distortion $p_2(j)$ can be represented as a Gaussian-weighted sinusoid, also known as a Gabor function. Eqs. 1 and 2 represent the lateral and axial PSFs where $\sigma_x$ represents the width of the transmitted ultrasonic beam, $\sigma_y$ is the pulse width of the ultrasonic wave that is transmitted through tissue, $f_0$ is its center frequency, and $c$ represents the speed of sound through tissue (Guha et al., 2025).

$$p_1(i) = \exp\left(-\frac{i^2}{2\sigma_x^2}\right) \tag{1}$$

$$p_2(j) = \sin\left(\frac{2\pi f_0}{c} j\right) \exp\left(-\frac{j^2}{2\sigma_y^2}\right) \tag{2}$$

The PSF is defined over spatial coordinates $i, j$, and $\sigma_x$ and $\sigma_y$ control the extent of blur induced by the PSF in both directions. Analytically, the PSF can be modeled as a filter. For a chosen size, the lateral and axial distortions can be represented as functions of $\sigma_x$ and $\sigma_y$, and the PSF is given by:

$$p(\sigma_x, \sigma_y) = p_1(\sigma_x) * p_2(\sigma_y) \tag{3}$$

A noisy ultrasound image $I_{noisy}$ corrupted by both additive ($\eta^a$) and multiplicative ($\eta^m$) noise and further degraded by the PSF can be represented by Eq. 4. Fig. 1 illustrates Eq. 4.

$$I_{noisy} = (I + I\,\eta^m) * p_1(\sigma_x) * p_2(\sigma_y) + \eta^a \tag{4}$$

## 4 Method

In this section, we show the forward and reverse processes for DEMIX, the training objective, and the model architecture.

### 4.1 Mixed Noise Formulations

The observed image $I_0$ is corrupted by additive and multiplicative noise and further degraded by the PSF $p$ of the imaging system. We consider multiple noise levels, and for a given noise level $t$, the noisy image is represented as $I_t$. The additive and multiplicative noise components are represented by $\eta_t^a$ and $\eta_t^m$, where $t \in \{1, 2, ..., T\}$, and $T$ is the maximum noise level.

The additive noise, arising from the thermal fluctuations in the sensors, is modeled as zero-mean Gaussian noise (Michailovich & Tannenbaum, 2006), and the speckle can be approximated as a zero-mean Gaussian with some variance. For a specific noise level $t$, additive noise (variance $\beta_t^2$) and speckle (variance $\alpha_t^2$) can be represented as $\eta_t^a$ and $\eta_t^m$, respectively.

$$\eta_t^m \sim \mathcal{N}(\mathbf{0}, \boldsymbol{\alpha_t^2}); \ \eta_t^a \sim \mathcal{N}(\mathbf{0}, \boldsymbol{\beta_t^2}) \tag{5}$$

Following Eq. 5, the noisy image $I_t$ can be expressed as Eq. 6.

$$I_t = I_0(1 + \eta_t^m) * p + \eta_t^a \tag{6}$$

For $\epsilon_a, \epsilon_m \sim \mathcal{N}(\mathbf{0}, \mathbf{I})$, the noisy image $I_t$ can be represented by Eq. 7.

$$I_t = I_0(1 + \alpha_t \epsilon_m) * p + \beta_t \epsilon_a \tag{7}$$

Assuming both additive and multiplicative noises vary linearly, $I_t$ can be expressed as Eq. 8 where $\alpha_t - \alpha_{t-1} = \delta$ and $\beta_t - \beta_{t-1} = \gamma$ $\forall t \geq 2$ and $t \leq T$.

$$I_t = I_0 * p + (I_0 \alpha_{t-1} \epsilon_m) * p + \beta_{t-1} \epsilon_a + (I_0 \delta \epsilon_m) * p + \gamma \epsilon_a \tag{8}$$

Next, we express $I_t$ in terms of $I_{t-1}$. Comparing Eq. 7 and Eq. 8, we obtain

$$I_t = I_{t-1} + (I_0 \delta \epsilon_m) * p + \gamma \epsilon_a \tag{9}$$

In terms of the initial image $I_0$, the noisy image $I_t$ can be expressed as follows, where $\epsilon_a, \epsilon_m \sim \mathcal{N}(\mathbf{0}, \mathbf{I})$ are randomly sampled.

$$I_t = I_0 + t[(I_0 \delta \epsilon_m) * p + \gamma \epsilon_a] \tag{10}$$

**Forward Process:** $\delta$ and $\gamma$ are positive scalars. $\epsilon_m \sim \mathcal{N}(\mathbf{0}, \mathbf{I})$, and $I_0 \epsilon_m$ represents element-wise multiplication of $I_0$ and $\epsilon_m$. The system PSF $p$ is the convolution of a zero-mean Gaussian function and a Gabor function. In the Fourier domain, this corresponds to a multiplication of the Fourier transforms of these functions and is analogous to a low-pass filter. When the frequency of the sinusoidal component in $p$ is high, the high-frequency components are attenuated and the resulting PSF $p$ can be approximated as a zero-mean Gaussian. Thus, in Eqs. 9 and 10, $(I_0 \delta \epsilon_m) * p = \delta (I_0 \epsilon_m) * p$ can be represented as $\delta K \epsilon'$, where $K$ is a function of the initial image $I_0$ and $\epsilon' \sim \mathcal{N}(\mathbf{0}, \mathbf{I})$. Now, $(I_0 \delta \epsilon_m) * p + \gamma \epsilon_a$ can be expressed as $\delta K \epsilon' + \gamma \epsilon_a$, where $\epsilon', \epsilon_a \sim \mathcal{N}(\mathbf{0}, \mathbf{I})$. Thus,

$$(I_0 \delta \epsilon_m) * p + \gamma \epsilon_a \sim \mathcal{N}(\mathbf{0}, (\delta^2 K^2 + \gamma^2)\mathbf{I}) \tag{11}$$

Assuming $(I_0 \delta \epsilon_m) * p + \gamma \epsilon_a$ is Gaussian (Eq. 11), Eq. 9 and Eq. 10 can be expressed as the following where $\epsilon \sim \mathcal{N}(\mathbf{0}, \mathbf{I})$.

$$I_t = I_{t-1} + \sqrt{\delta^2 K^2 + \gamma^2} \epsilon \tag{12}$$

$$I_t = I_0 + t\sqrt{\delta^2 K^2 + \gamma^2} \epsilon \tag{13}$$

Thus, the forward process is given by Eq. 14 and Eq. 15.

$$q(I_t|I_{t-1}) \sim \mathcal{N}(I_t; I_{t-1}, (\delta^2 K^2 + \gamma^2)\mathbf{I}) \tag{14}$$

$$q(I_t|I_0) \sim \mathcal{N}(I_t; I_0, t^2(\delta^2 K^2 + \gamma^2)\mathbf{I}) \tag{15}$$

**Reverse Process:** To design the reverse process, the analytical inverse of the forward process $q(I_{t-1}|I_t)$ is calculated.

$$q(I_{t-1}|I_0, I_t) \sim \mathcal{N}(I_{t-1}; \mu_q(I_t, I_0), \Sigma_q(I_0, t)) \tag{16}$$

In Eq. 16, $\mu_q(I_t, I_0) = \dfrac{(t-1)^2 I_t + I_0}{(t-1)^2 + 1}$ and $\Sigma_q(I_0, t) = \dfrac{(\delta^2 K^2 + \gamma^2)(t-1)^2}{(t-1)^2 + 1}$. Thus, the reverse process can be formulated as

$$f_\theta(I_{t-1}|I_t) \sim \mathcal{N}(I_t; \mu_\theta(I_t, \alpha, \beta, \psi), \Sigma_\theta(I_t, t)) \tag{17}$$

In Eq. 17, $\sigma$, $\alpha$, and $\psi$ represent the additive noise schedule, the multiplicative noise schedule, and the PSF parameters, respectively. In the reverse process, we assume $\Sigma_\theta(I_t, t) = \Sigma_q(I_0, t)$.

$$\mu_\theta(I_t, \alpha, \beta, \psi) = \frac{(t-1)^2 I_t + I_\theta(I_t, \alpha, \beta, \psi)}{(t-1)^2 + 1} \tag{18}$$

**Training Objective** In the reverse process, $\mu_\theta(I_t, \alpha, \beta, \psi)$ and $\Sigma_\theta(I_t, t)$ are functions of the noisy image $I_t$. The first component of the loss function for DEMIX is:

$$\mathcal{L}_{\mathrm{D}} = \mathbf{E}[\|I_\theta(I_t, \alpha, \beta, \psi) - I_0\|_1] \tag{19}$$

The diffusion loss is complemented with the multiscale SSIM loss to enhance the quality of the reconstructed images.

$$\mathcal{L}_{\mathrm{MS\text{-}SSIM}} = 1 - \mathrm{MS\text{-}SSIM}(I_\theta(I_t, \alpha, \beta, \psi), I_0) \tag{20}$$

The overall loss function is expressed as

$$\mathcal{L} = \mathcal{L}_{\mathrm{D}} + \mathcal{L}_{\mathrm{MS\text{-}SSIM}} \tag{21}$$

## 4.2 Model Architecture

In this section, we present a novel dual encoder framework for denoising images that are simultaneously corrupted with additive and multiplicative noise and further degraded by the PSF of the imaging system. In this framework, one encoder is dedicated to noise, and the other to the PSF. The model explicitly accounts for the axial and lateral distortions of the PSF, enabling adaptive restoration of noisy images while preserving critical structural details. The key components of the proposed model are:

1. The network can reduce arbitrary levels of additive and multiplicative noise along with PSF distortions along the axial and lateral directions in a single step.

2. Each encoder independently encodes the additive and multiplicative noise characteristics. Their latent representations are integrated with a masked gating mechanism to feed the bottleneck layer and the skip connections, effectively disentangling the noise components.

3. Additive and multiplicative noise schedules, $\sigma$ and $\alpha$, are embedded within each convolutional block, enabling the model to dynamically assess noise intensity and achieve robust denoising across images corrupted with varying noise intensities without explicit noise supervision.

4. The complete spectrum of axial and lateral PSF distortions ($\psi$) is encoded and integrated into the encoder layers. This enables the model to dynamically estimate the degree of structural degradation, leading to an effective restoration of clean images.

An overview of the proposed architecture is given in Fig. 2. The different components are described below. The complete code will be published for reproducibility.

---

**Algorithm 1** Training DEMIX

**Input**: $\{I\}, \alpha, \beta, \psi, I_\theta$
**Output**: Trained $I_\theta$

1:  **while** not converged **do**
2:      $I_0 \in \{I\}$
3:      $\sigma_x, \sigma_y \sim \psi$
4:      $t \sim \text{Uniform}(1, ..., T)$
5:      $\epsilon_m \sim \mathcal{N}(\mathbf{0}, \mathbf{I}), \ \epsilon_a \sim \mathcal{N}(\mathbf{0}, \mathbf{I})$
6:      $I_t = I_0(1 + \alpha_t \epsilon_m) * p(\sigma_x, \sigma_y) + \beta_t \epsilon_a$
7:      $\tilde{I} = x_\theta(I_t, \alpha, \beta, \psi)$
8:      Compute $\nabla_\theta \|\tilde{I} - I_0\|_1$
9:      Update $\theta$
10: **end while**

---

**Noise Encoder:** The noise encoder integrates the additive and multiplicative noise schedules, $\beta \in \mathbb{R}^{1 \times T}$ and $\alpha \in \mathbb{R}^{1 \times T}$, into a unified embedding that quantifies noise severity without explicit supervision. For a noisy $n \times n$ sized input image, the noise schedules are first projected to a high-dimensional space, $\beta_{pos} \in \mathbb{R}^{T \times n}$ and $\alpha_{pos} \in \mathbb{R}^{T \times n}$ with sinusoidal positional embeddings and thereafter processed with a two-layer MLP with GELU activation to obtain $\beta'_{pos} \in \mathbb{R}^{T \times 4n}$ and $\alpha'_{pos} \in \mathbb{R}^{T \times 4n}$. The learned embeddings $\beta'_{pos}$ and $\alpha'_{pos}$ are processed through parallel MLP layers to obtain intermediate representations $z_\beta$ and $z_\alpha$, where $z_\beta = ReLU(W_\beta \beta'_{pos})$ and $z_\alpha = ReLU(W_\alpha \alpha'_{pos})$. These are added, and a final MLP layer is applied to fuse the noise characteristics $\widetilde{\alpha\beta} = ReLU(W_\beta(z_\beta + z_\alpha)) \in \mathbb{R}^{T \times 4n}$.

**PSF Encoder:** The PSF encoder models the spatial degradations caused by the axial and lateral distortions of the ultrasound imaging system. For each direction, the minimum and maximum distortions are chosen. Next, a parameter vector $\in \mathbb{R}^{1 \times m}$ is constructed by interpolating $m$ values between the chosen minimum and maximum values of distortions and stacked to obtain $\psi \in \mathbb{R}^{2 \times m}$. $\psi$ captures the complete spectrum of PSF degradations and facilitates generalization in varying degrees of spatial degradation. $\psi$ is first embedded with sinusoidal positional embeddings to obtain high dimensional embedding $\psi_{pos} \in \mathbb{R}^{2 \times 1 \times P_d}$, where $P_d = 2m$. The positional embedding is processed through a two-layer MLP with the GELU activation function, resulting in the final PSF embedding $\Psi = MLP(\psi_{pos}) \in \mathbb{R}^{2 \times 1 \times m}$. This embedding enables the model to dynamically adapt to varying degrees of PSF distortion for effective image reconstruction.

**Gated Fusion Block:** The Gated Fusion Block integrates the latent features of the multiplicative and additive noise encoders $\chi_{mul}$ and $\chi_{add}$. This strategy is similar to the gating mechanism proposed in (Hu

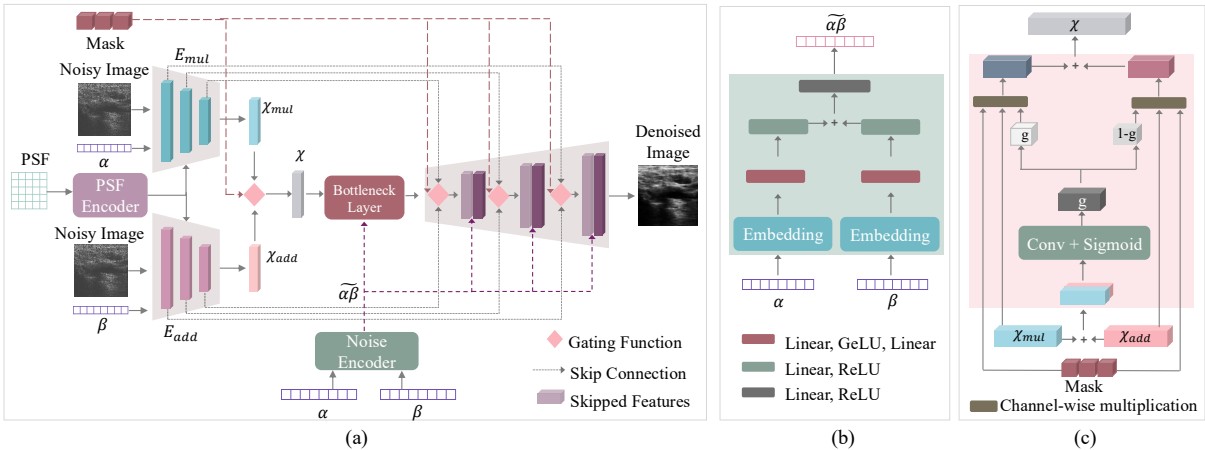

Figure 2: (a) The **overall architecture** is shown and $\alpha$ and $\beta$ are the multiplicative and additive schedules. The dual encoder architecture processes and extracts the additive and multiplicative noise features separately and combines them through the gated fusion block. The combined features are subsequently processed by the decoder layers, and the final denoised image is obtained. (b) The **noise encoder** combines the multiplicative and additive noise schedules ($\alpha$ and $\beta$, respectively) and embeds them together to create the final noise embedding vector $\widetilde{\alpha\beta}$, which is combined with the bottleneck layer and all decoder layers of DEMIX. (c) The **gated fusion block** combines the multiplicative $\chi_{mul}$ and additive $\chi_{add}$ noise features, along with the sampled mask and fuses them to obtain a feature vector $\chi$ for further processing in the bottleneck layer and the decoder layers for efficient denoising performance.

et al., 2018). To promote robust training, a masking strategy ensures that at least one of the encoders remains active, with the same masking strategy applied to both the bottleneck layer and the skip connections. Let $f_1, f_2$ represent multiplicative and additive features for different encoder layers, respectively. For features $f_1, f_2 \in \mathbb{R}^{B \times C \times H \times W}$, the features are fused through the learned gate $g = \phi(Conv_{1 \times 1}([f_1, f_2])) \in \mathbb{R}^{B \times C \times H \times W}$, where $[f_1, f_2]$ represents channelwise concatenation, $Conv_{1 \times 1}$ represents convolution by a $1 \times 1$ kernel and $\phi$ is the Sigmoid activation function. The learned gate $g$ determines the influence of each encoder on the final embedding. A mask $m = \{[1, 1], [1, 0], [0, 1]\}$ is randomly sampled for each image, ensuring that at least one encoder contributes to the final learned representations. The final fused representation is given by $f = m_1 \circ g \circ f_1 + m_2 \circ (1 - g) \circ f_2$ where $m_1$ and $m_2$ are the broadcast masks and $\circ$ represents element-wise multiplication.

**Denoising Framework:** The reverse process is implemented with a dual encoder UNet backbone and a masked gated fusion block that unifies the additive $\chi_{add}$ and multiplicative $\chi_{mul}$ features for bottleneck and skip connections. Each encoder processes the input images through multiple ConvNeXt-like residual blocks, followed by attention blocks and downsampling mechanisms. To enhance robustness, one encoder is randomly masked during training, forcing the network to learn semantically meaningful features even from partial information. The noise encoder encodes both $\beta$ and $\alpha$ in a unified representation $\widetilde{\alpha\beta}$, which conditions the bottleneck layer and all the decoder layers. Additionally, each encoder layer is integrated with the PSF features $\Psi$, allowing the model to account for spatial degradations. The decoder comprises multiple upsampling blocks. At each decoder level, the encoder representations are integrated with the decoder representations and processed through residual blocks and attention modules. The overall training methodology is given in Algorithm 1. The additive and multiplicative noise schedules, along with lateral and axial PSF distortions, are encoded within the framework. To enforce robustness, masks $\in \{[1, 1], [1, 0], [0, 1]\}$ are randomly sampled. For [1, 0] and [0, 1], the additive and multiplicative components are dropped, respectively. This ensures that DEMIX learns to disentangle the individual noise components and adaptively denoise different intensities of mixed noise along with varying degrees of PSF degradation, without any additional supervision.

### 4.3 Training Methodology

During training, for each batch, the lateral and axial distortions, $\sigma_x$ and $\sigma_y$ are sampled from $\psi$ and the noise level is sampled as $t \sim \text{Uniform}(1, 2, ..., T)$. Using $\epsilon_m \sim \mathcal{N}(\mathbf{0}, \mathbf{I})$ and $\epsilon_a \sim \mathcal{N}(\mathbf{0}, \mathbf{I})$, the noisy batch $I_t$ is obtained following Eq. 7 and passed through the neural network $I_\theta$ to obtain denoised images. This ensures the model learns to adaptively denoise different intensities of mixed noise and varying degrees of PSF degradation.

### 4.4 Datasets

**Ultrasound Dataset:** This dataset comprises two different classes of ultrasound images acquired from human subjects. The first group contains images of the brachial plexus, lymph nodes and fetal heads (Kaggle, 2016). The second group consists of images of the common carotid artery (CCA) (Momot, 2022). The images of the first group are $256 \times 320$ pixels. The size of images in the second category range between $230 \times 390$ pixels and $450 \times 600$ pixels.

**Phantom Dataset:** This dataset comprises images of eight categories, with each category comprising 100 images. For six categories of the acquired images, each group has at least axial-lateral resolution targets or two different grayscale targets. The other two categories were acquired at different depths (40 mm and 70 mm), introducing different degrees of ultrasound attenuation (Guha et al., 2025).

Table 1: Quantitative comparisons for different denoising algorithms on different datasets. The best value is presented in bold while the second best is underlined. Higher values are expected for both PSNR and SSIM.

| Methods | $\alpha_t = 0.373$ $\beta_t = 0.127$ PSNR | SSIM | $\alpha_t = 0.749$ $\beta_t = 0.251$ PSNR | SSIM | $\alpha_t = 1.124$ $\beta_t = 0.376$ PSNR | SSIM | $\alpha_t = 1.5$ $\beta_t = 0.5$ PSNR | SSIM | $\alpha_t = 0.373$ $\beta_t = 0.127$ PSNR | SSIM | $\alpha_t = 0.749$ $\beta_t = 0.251$ PSNR | SSIM | $\alpha_t = 1.124$ $\beta_t = 0.376$ PSNR | SSIM | $\alpha_t = 1.5$ $\beta_t = 0.5$ PSNR | SSIM |
|---|---|---|---|---|---|---|---|---|---|---|---|---|---|---|---|---|
| | \multicolumn — $\sigma_x = 2, \sigma_y = 1.5$ | | | | | | | | \multicolumn — $\sigma_x = 4, \sigma_y = 3.5$ | | | | | | | |
| ULTRASOUND DATASET | | | | | | | | | | | | | | | | |
| BM3D (Dabov et al., 2007) | 26.31 | 0.672 | 19.12 | 0.116 | 15.73 | 0.033 | 14.67 | 0.015 | 26.35 | 0.672 | 19.28 | 0.117 | 15.85 | 0.033 | 14.83 | 0.015 |
| NLMeans (Buades et al., 2005) | 25.42 | 0.478 | 22.81 | 0.250 | 22.17 | 0.146 | 20.76 | 0.095 | 25.33 | 0.478 | 22.76 | 0.249 | 21.24 | 0.148 | 21.00 | 0.096 |
| DnCNN (Zhang et al., 2017) | 19.55 | 0.238 | 20.39 | 0.212 | 20.44 | 0.188 | 20.74 | 0.177 | 19.77 | 0.241 | 20.20 | 0.211 | 20.50 | 0.190 | 20.18 | 0.176 |
| SCUNet (Zhang et al., 2023) | 5.80 | 0.113 | 6.56 | 0.092 | 7.38 | 0.070 | 8.47 | 0.051 | 5.79 | 0.113 | 6.51 | 0.092 | 7.41 | 0.071 | 8.84 | 0.051 |
| SwinIR (Liang et al., 2021) | 5.35 | 0.154 | 6.63 | 0.109 | 7.88 | 0.071 | 9.48 | 0.044 | 5.42 | 0.154 | 6.63 | 0.109 | 7.76 | 0.071 | 9.51 | 0.044 |
| Pureformer (Gautam et al., 2025) | 5.96 | 0.142 | 7.08 | 0.101 | 8.51 | 0.067 | 11.22 | 0.042 | 5.94 | 0.142 | 7.24 | 0.101 | 9.06 | 0.067 | 11.20 | 0.042 |
| AdaReNet (Liu et al., 2025) | 6.98 | 0.111 | 7.74 | 0.080 | 8.69 | 0.058 | 9.92 | 0.040 | 6.91 | 0.111 | 7.77 | 0.080 | 8.96 | 0.057 | 9.91 | 0.040 |
| SDDPM (Guha & Acton, 2023) | 26.16 | 0.727 | 23.78 | 0.643 | 22.47 | 0.590 | 21.58 | 0.535 | 26.24 | 0.727 | 23.60 | 0.646 | 22.62 | 0.588 | 21.61 | 0.545 |
| PSF-SRDN (Guha et al., 2025) | 25.96 | 0.723 | 23.92 | 0.647 | 22.83 | 0.592 | 21.88 | 0.556 | 26.06 | 0.722 | 24.03 | 0.650 | 22.71 | 0.595 | 21.85 | 0.556 |
| **DEMIX** (Ours) | **26.85** | **0.745** | **24.55** | **0.671** | **23.22** | **0.629** | **22.27** | **0.586** | **26.81** | **0.747** | **24.36** | **0.675** | **22.95** | **0.629** | **22.24** | **0.585** |
| PHANTOM DATASET | | | | | | | | | | | | | | | | |
| BM3D (Dabov et al., 2007) | 24.33 | 0.637 | 16.78 | 0.087 | 14.57 | 0.022 | 13.88 | 0.010 | 24.20 | 0.637 | 16.74 | 0.087 | 14.56 | 0.022 | 13.89 | 0.010 |
| NLMeans (Buades et al., 2005) | 22.91 | 0.442 | 19.89 | 0.245 | 19.67 | 0.148 | 18.18 | 0.096 | 23.34 | 0.472 | 20.02 | 0.242 | 18.80 | 0.138 | 18.17 | 0.086 |
| DnCNN (Zhang et al., 2017) | 19.87 | 0.224 | 19.96 | 0.188 | 19.80 | 0.180 | 19.73 | 0.187 | 19.95 | 0.225 | 20.12 | 0.187 | 19.63 | 0.180 | 19.85 | 0.188 |
| SCUNet (Zhang et al., 2023) | 7.08 | 0.158 | 7.62 | 0.142 | 8.39 | 0.120 | 9.49 | 0.095 | 7.08 | 0.158 | 7.63 | 0.142 | 8.39 | 0.121 | 9.47 | 0.095 |
| SwinIR (Liang et al., 2021) | 5.02 | 0.244 | 6.11 | 0.160 | 7.53 | 0.093 | 9.75 | 0.051 | 5.02 | 0.243 | 6.10 | 0.160 | 7.65 | 0.092 | 9.68 | 0.051 |
| Pureformer (Gautam et al., 2025) | 5.81 | 0.190 | 7.91 | 0.133 | 10.65 | 0.081 | 13.38 | 0.047 | 5.74 | 0.190 | 7.95 | 0.133 | 10.59 | 0.082 | 13.20 | 0.047 |
| AdaReNet (Liu et al., 2025) | 5.26 | 0.179 | 5.68 | 0.126 | 6.18 | 0.081 | 6.96 | 0.050 | 5.27 | 0.179 | 5.69 | 0.126 | 6.17 | 0.081 | 6.94 | 0.050 |
| SDDPM (Guha & Acton, 2023) | 22.96 | 0.652 | 21.06 | 0.600 | 20.06 | 0.584 | 19.31 | 0.577 | 22.94 | 0.652 | 21.08 | 0.600 | 19.98 | 0.584 | 19.31 | 0.577 |
| PSF-SRDN (Guha et al., 2025) | 22.14 | 0.636 | 20.79 | 0.603 | 19.79 | 0.583 | 19.06 | 0.574 | 22.14 | 0.636 | 20.76 | 0.603 | 19.74 | 0.583 | 19.09 | 0.569 |
| **DEMIX** (Ours) | **25.10** | **0.664** | **22.33** | **0.607** | **21.29** | **0.589** | **20.76** | **0.581** | **25.00** | **0.664** | **22.41** | **0.607** | **21.38** | **0.589** | **20.79** | **0.581** |

## 5 Experiments

### 5.1 Implementation Details

All experiments were implemented on a NVIDIA RTX 3090 GPU. The stochastic gradient descent (SGD) optimizer and a batch size of 128 were used. All models were trained on randomly cropped $64 \times 64$ patches of noisy, PSF-distorted images for 150 epochs for the ultrasound dataset and 200 epochs for the phantom dataset. The experiments were trained with an initial learning rate of 0.005, which was reduced to 10% of the value every 50 epochs. For both datasets, three randomly sampled training and validation partitions are used. Each time, 70% of the images were reserved for training and the rest 30% were used to validate the trained model. The images in the training partition were randomly augmented by rotation and Gaussian noise having $\mu = 0.025, 0.05$ and $\sigma^2 = 0, 0.001$. The augmented training set contains approximately 6000 images and 13000 images for the ultrasound and phantom datasets, respectively. The proposed mixed noise denoising model was trained with $\alpha = [0.005, 1.5]$, $\beta = [0.005, 0.5]$ and $T = 200$. A $3 \times 3$ PSF was considered for the ultrasound imaging system that has a center frequency $f_0 = 10\,\text{MHz}$ and the speed of sound through

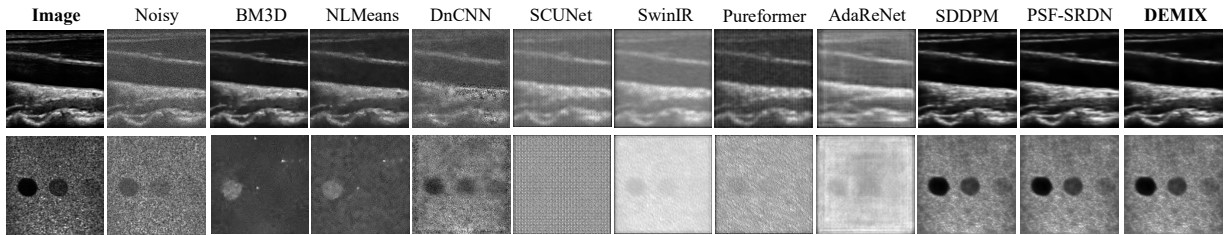

Figure 3: Qualitative comparisons of different denoising methods for $\alpha_t = 0.373, \beta_t = 0.127$ and $\sigma_x = 3, \sigma_y = 2.5$. Column 1: Ground truth images from ultrasound dataset (row 1) and the phantom dataset (row 2). Column 2: Noisy images. Columns 3 - 12: Denoised images by different algorithms. The results for the proposed method DEMIX are shown in the last column.

the tissue is $c = 1540\,\mathrm{m/s}$. The lateral and axial distortions for the ultrasound PSF were bounded between $\sigma_x = [1, 4]$ and $\sigma_y = [0.5, 3.5]$. 50 values of ($m = 50$) were interpolated for both $\sigma_x$ and $\sigma_y$ to facilitate a detailed evaluation over a wide range of axial and lateral distortions.

### 5.2 Comparisons with the State-of-the-Art

**Evaluation Metrics:** We report the peak signal-to-noise ratio (PSNR) and the structural similarity index measure (SSIM) for all experiments. PSNR, reported in decibels (dB), quantifies the intensity differences between the reconstructed and ground truth images, and SSIM evaluates perceptual quality by assessing luminance, contrast, and structural consistency. Higher values of PSNR and SSIM indicate better results.

Since DEMIX is proposed for image enhancement and not for generation purposes, we focus on distortion-based evaluation metrics such as PSNR and SSIM that measure the quality of the restored images compared to the true underlying signal. In this work, we are not comparing a generated image to a real image. Hence, the distributional distance obtained with FID may not be an effective judge of enhancement.

**Baseline Methods:** We conducted extensive experiments on the two datasets to compare the performance of DEMIX with the state-of-the-art image denoising methods, including BM3D (Dabov et al., 2007), NLMeans (Buades et al., 2005), DnCNN (Zhang et al., 2017), SwinIR (Liang et al., 2021), SCUNet (Zhang et al., 2023), Pureformer (Gautam et al., 2025), AdaReNet (Liu et al., 2025), SDDPM (Guha & Acton, 2023) and PSF-SRDN (Guha et al., 2025). BM3D (Dabov et al., 2007) and NLMeans (Buades et al., 2005) are traditional image denoising algorithms, whereas all other baselines are data-driven. BM3D (Dabov et al., 2007) denoises by grouping similar patches into 3D stacks and collaboratively filtering them in a transform domain, whereas NLMeans (Buades et al., 2005) denoises by averaging the pixel intensities having similar local neighborhoods throughout the image. DnCNN (Zhang et al., 2017) targets Gaussian denoising by learning image residuals to separate clean latent images from noisy inputs. SwinIR (Liang et al., 2021) combines Swin transformer (Liu et al., 2021) layers and residual connections for an image restoration algorithm and integrates shallow and deep feature extractions with high fidelity image reconstruction. SCUNet (Zhang et al., 2023) integrates new swin-conv blocks into a UNet backbone for a blind image denoising framework. Pureformer (Gautam et al., 2025) is an encoder-decoder architecture with transformer-based skip connections and comprises latent feature enhancer blocks that use a spatial filter bank to expand the receptive field, thereby improving image restoration. AdaReNet (Liu et al., 2025) proposes a rotation-invariant prior in a supervised learning framework, and rotation-invariant convolutional layers are used to replace all translation-invariant convolutional layers in a UNet backbone. SDDPM (Guha & Acton, 2023) and PSF-SRDN (Guha et al., 2025) are diffusion models specifically proposed to reduce signal-dependent speckle. While SDDPM is designed to reduce speckle, PSF-SRDN is additionally equipped with the PSF information that helps to reduce speckle and correct PSF-induced distortions.

**Results:** All methods were evaluated for different strengths of additive and multiplicative noise for varying degrees of PSF distortions. To systematically evaluate DEMIX for different degradation levels, degraded images were synthetically obtained following the forward process. The evaluation scores for the two datasets are demonstrated in Table 1. All baseline methods were evaluated on same batch size, learning rate, optimizer

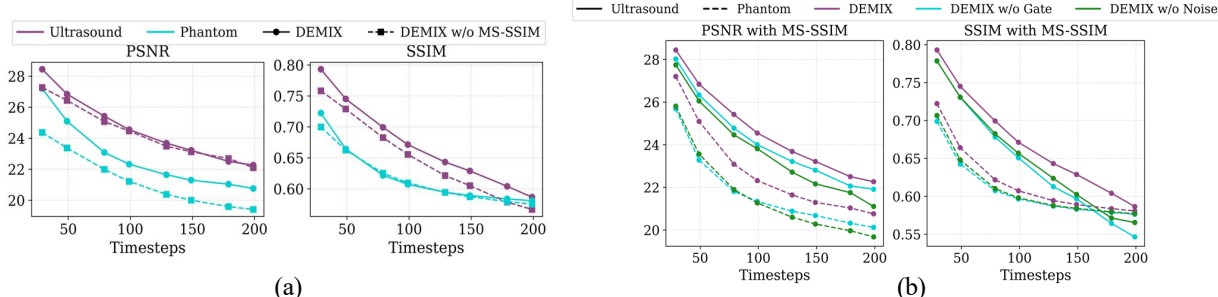

Figure 4: (a) The MS-SSIM loss (shown for $\sigma_x = 2, \sigma_y = 1.5$) improves the PSNR and SSIM scores for both datasets for the proposed method DEMIX. (b) With MS-SSIM loss, the PSNR and SSIM scores (shown for $\sigma_x = 2, \sigma_y = 1.5$) for DEMIX are consistently higher than the ablated models without the gated fusion and the noise encoder.

Table 2: Ablation studies of different components of DEMIX on Ultrasound and Phantom datasets. PSNR and SSIM are reported for $\sigma_x = 2, \sigma_y = 1.5$. The best scores are shown in bold letters.

| Dataset | Method | Loss | $\alpha_t = 0.373$ $\beta_t = 0.127$ PSNR | SSIM | $\alpha_t = 0.749$ $\beta_t = 0.251$ PSNR | SSIM | $\alpha_t = 1.124$ $\beta_t = 0.376$ PSNR | SSIM | $\alpha_t = 1.5$ $\beta_t = 0.5$ PSNR | SSIM |
|---|---|---|---|---|---|---|---|---|---|---|
| Ultrasound | DEMIX w/o Noise Encoder | $\mathcal{L}_D$ | 26.08 | 0.729 | 23.85 | 0.654 | 22.53 | 0.601 | 21.44 | 0.557 |
| | DEMIX w/o Noise Encoder | $\mathcal{L}_D + \mathcal{L}_{MS\text{-}SSIM}$ | 26.05 | 0.731 | 23.82 | 0.657 | 22.17 | 0.602 | 21.10 | 0.565 |
| | DEMIX w/o Gated Fusion | $\mathcal{L}_D$ | 26.45 | 0.728 | 23.99 | 0.651 | 22.55 | 0.597 | 21.91 | 0.564 |
| | DEMIX w/o Gated Fusion | $\mathcal{L}_D + \mathcal{L}_{MS\text{-}SSIM}$ | 26.34 | 0.731 | 24.00 | 0.651 | 22.81 | 0.597 | 21.91 | 0.546 |
| | DEMIX | $\mathcal{L}_D$ | 26.45 | 0.730 | 24.46 | 0.657 | 22.96 | 0.603 | 22.46 | 0.567 |
| | DEMIX (single encoder) | $\mathcal{L}_D + \mathcal{L}_{MS\text{-}SSIM}$ | 25.43 | 0.708 | 22.94 | 0.624 | 21.78 | 0.567 | 20.91 | 0.528 |
| | **DEMIX (proposed)** | $\mathcal{L}_D + \mathcal{L}_{MS\text{-}SSIM}$ | **26.85** | **0.745** | **24.55** | **0.671** | **23.22** | **0.629** | **22.27** | **0.586** |
| Phantom | DEMIX w/o Noise Encoder | $\mathcal{L}_D$ | 24.17 | 0.650 | 21.96 | 0.600 | 20.94 | 0.586 | 20.25 | 0.578 |
| | DEMIX w/o Noise Encoder | $\mathcal{L}_D + \mathcal{L}_{MS\text{-}SSIM}$ | 23.56 | 0.648 | 21.27 | 0.598 | 20.28 | 0.583 | 19.68 | 0.577 |
| | DEMIX w/o Gated Fusion | $\mathcal{L}_D$ | 23.12 | 0.642 | 21.20 | 0.597 | 20.36 | 0.583 | 19.79 | 0.577 |
| | DEMIX w/o Gated Fusion | $\mathcal{L}_D + \mathcal{L}_{MS\text{-}SSIM}$ | 23.28 | 0.643 | 21.34 | 0.597 | 20.68 | 0.583 | 20.13 | 0.576 |
| | DEMIX | $\mathcal{L}_D$ | 23.37 | 0.663 | 21.21 | **0.609** | 20.00 | 0.587 | 19.40 | 0.575 |
| | DEMIX (single encoder) | $\mathcal{L}_D + \mathcal{L}_{MS\text{-}SSIM}$ | 22.52 | 0.617 | 21.19 | 0.586 | 20.74 | 0.577 | 20.13 | 0.571 |
| | **DEMIX (proposed)** | $\mathcal{L}_D + \mathcal{L}_{MS\text{-}SSIM}$ | **25.10** | **0.664** | **22.33** | 0.607 | **21.29** | **0.589** | **20.76** | **0.581** |

and step size as in Section 5.1. The batch size and step size were ablated as 8, 16, 32, 64, 128 and 10, 20, respectively. Since the resulting PSNR and SSIM scores did not vary significantly across these configurations, we report the scores obtained with hyperparameters specified in Section 5.1. The proposed method, DEMIX, outperforms all baseline methods for the ultrasound dataset and obtains competitive scores for the phantom dataset. DEMIX achieves the best SSIM scores for both datasets for all noise levels, outperforming most state-of-the-art methods by a significant margin. SDDPM and PSF-SRDN are specifically proposed for multiplicative noise and perform better than other baselines. Fig. 3 shows the original image, the noisy

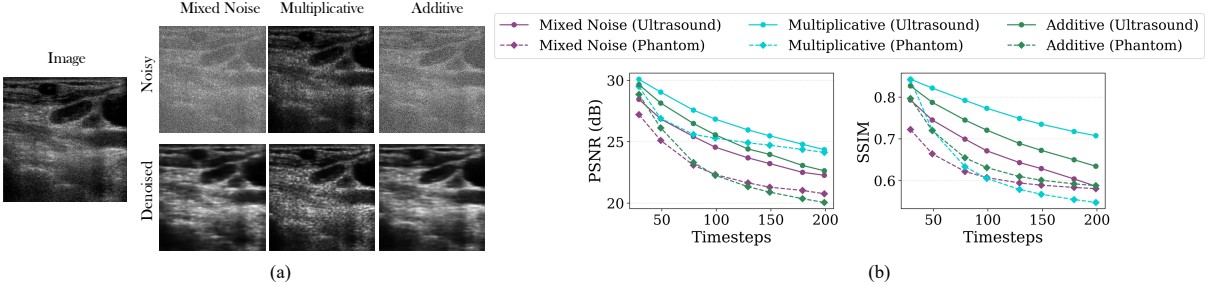

Figure 5: (a) $\sigma_x = 3, \sigma_y = 2.5$, $\alpha_t = 0.598$, $\beta_t = 0.201$ (b) For PSF parameters $\sigma_x = 2, \sigma_y = 1.5$, we show how the PSNR and SSIM scores vary for the different noise combinations for the two datasets. All other PSF distorted images follow similar trends.

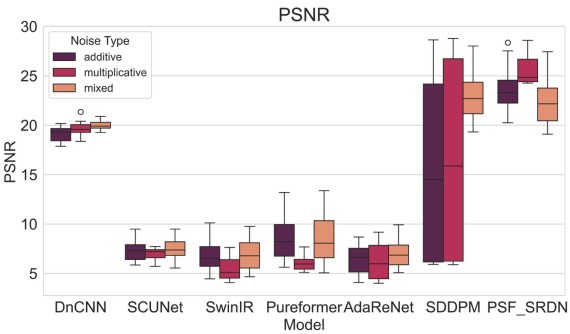 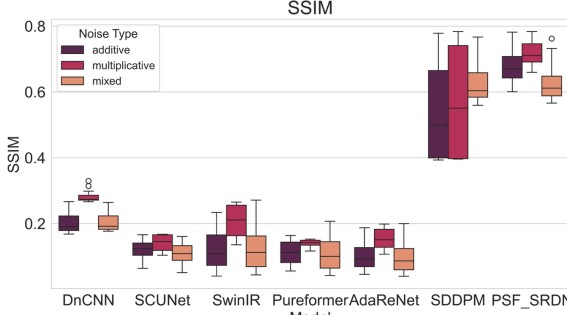

Figure 6: All baseline models were trained on mixed noise conditions. We evaluate the methods on three different noise conditions and the PSNR and SSIM scores are reported.

Table 3: Downstream evaluation: Segmentation scores for the original dataset and the dataset denoised by DEMIX. We report the Jaccard Similarity (JS) and the Dice Score (%) for an overlap of 0.5.

| Method | Original | | Denoised | |
|---|---|---|---|---|
| | JS | Dice (%) | JS | Dice(%) |
| UNet (Ronneberger et al., 2015) | 0.675 | 80.54 | **0.696** | **82.00** |
| AttnUNet (Oktay et al., 2018) | 0.681 | 80.89 | **0.691** | **81.60** |
| Segformer (Xie et al., 2021) | 0.337 | 50.27 | **0.593** | **74.01** |
| TransUNet (Chen et al., 2021) | **0.637** | 77.72 | 0.634 | **77.84** |

image, and the denoised images obtained from the state-of-the-art denoising algorithms. It can be seen that the images denoised by DEMIX recover the fine structural details present in the original image. Moreover, it can be verified from Table 1 that the PSNR and SSIM scores for the images denoised by DEMIX are significantly higher than all comparative methods even for low SNR and high noise scenarios. DEMIX, trained on mixed noise, is evaluated on individual degradations and the results are given in Fig. 5: (a) shows the denoised images reconstructed from different noisy inputs and (b) shows the PSNR and SSIM scores for the different degradation sources. In Fig. 6, we show how the performances of the different baseline methods vary for different noise combinations. The PSNR and SSIM scores for the additive, multiplicative and mixed noise conditions follow the trends observed in Table 1.

**Downstream Task:** We evaluated the performance of the denoised images for the downstream segmentation task. We used the thyroid nodule segmentation dataset, TN3K (Gong et al., 2021; 2022), which contains 3493 thyroid nodule images along with nodule mask labels. We have used UNet (Ronneberger et al., 2015), AttenUNet (Oktay et al., 2018), Segformer (Xie et al., 2021), and TransUNet (Chen et al., 2021) to evaluate the segmentation performance of the TN3K dataset for the original and denoised images, where the train and test partitions contained 2879 and 614 images, respectively. Table 3 shows the Jaccard Similarity (JS) and the Dice Scores (%). To evaluate the segmentation models with the denoised images, both the training and test partitions were denoised with the model pre-training on the ultrasound data set. It can be seen that, for the majority of the cases, the denoised images perform better, demonstrating the superiority of the proposed denoising method.

### 5.3 Ablation Studies

We study the effect of different components of the proposed architecture. Table 2 shows the PSNR and SSIM scores for different variations of DEMIX for $\sigma_x = 2, \sigma_y = 1.5$. All other combinations of PSF distortions demonstrate similar trends. It can be seen that the proposed architecture and the overall objective function ($\mathcal{L}_D + \mathcal{L}_{\text{MS-SSIM}}$) for DEMIX, including the noise encoder and the gating mechanism, yield the highest evaluation scores among all other variations. Fig. 4 (a) shows the improvement in the evaluation scores for

DEMIX with the inclusion of $\mathcal{L}_{\text{MS-SSIM}}$, while Fig. 4 (b) shows the effectiveness of the noise encoder and the gated fusion block in the final evaluation scores. The multiscale SSIM loss consistently improves the evaluation scores.

# 6 Discussions

## 6.1 Target modality for the proposed method

Even though DEMIX has primarily been evaluated for B-mode images, it can be applied to all coherent imaging modalities such as synthetic aperture radar (SAR), sonar, laser and electron microscopy images. When the transmitted ultrasonic waves are reflected from the different tissue boundaries, the phase and the amplitude for each wave change while the frequency remains constant. When the waves from a large number of reflectors are superimposed, the resulting expression can be expressed as the product of the true underlying signal and speckle (Goodman, 2007). Specifically for B-mode ultrasound images, the speckled signal represents the signature of the tissue microstructure and the system PSF governs the resolution (Cloutier et al., 2021). In other coherent imaging systems, such as X-ray microtomography, the acquired signal is defined as a speckled signal convolved with the intensity PSF (Lee et al., 2026). Additionally, all digital images are affected by signal-independent noise, which gets introduced in different stages of acquisition and quantization processes (Gonzalez, 2009). Thus, DEMIX provides a principled framework for robust image enhancement for diverse coherent imaging modalities.

## 6.2 Generalizability of the proposed model

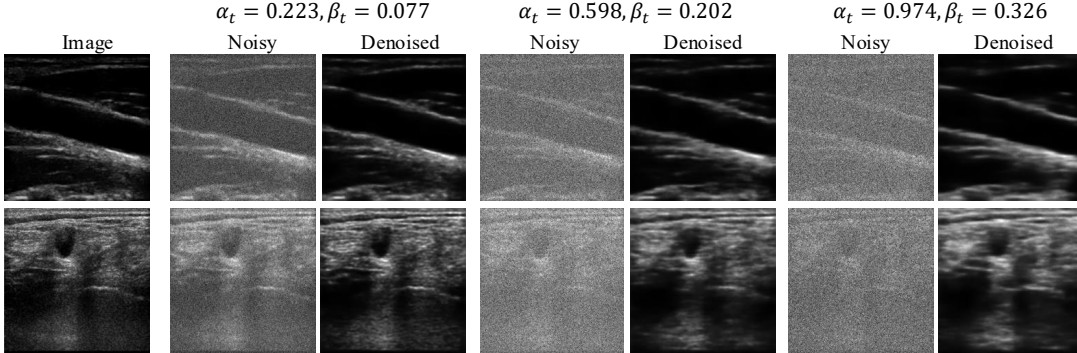

Figure 7: Each row shows the original image and the noisy and denoised images for different combinations of additive and multiplicative noise.

DEMIX is trained on degraded inputs having speckle, additive noise and PSF-induced distortions. Fig 5 demonstrates the denoising performance for individual degradations during inference. DEMIX is generalizable, does not overfit to the joint distribution of the additive and multiplicative noise components and is well-suited for real-world denoising, where the PSF distortion parameters or the exact noise combinations are not explicitly known. In real-world image denoising scenarios, either of the noise components can be absent, and this might lead to undesired performances from the denoising model. Fig. 7 demonstrates the denoising performance for different levels of additive and multiplicative noise components, where DEMIX recovers the high-frequency details without over-smoothing the edges. Additionally, the entire noise schedules $\alpha$ and $\beta$, along with the entire range of PSF distortions $\psi$, are encoded in the architecture. DEMIX does not require the exact noise level or the PSF distortion to be known during inference, which makes it suitable for real-world applications.

## 6.3 Efficient training

DEMIX is essentially an image denoising framework proposed in an inverse problem setting. Instead of using the full resolution images, the model can be trained on randomly cropped, small patches that capture the

representative noise patterns. As a result, the training complexity is significantly reduced. The final model can be generalized to all image sizes. After training, DEMIX can be used for denoising images of arbitrary sizes, without any additional fine-tuning or retraining. Essentially, DEMIX learns the noise characteristics from image patches, which is seamlessly transferred to full-sized images during inference. Patch-efficient training, combined with the formulation of inverse problems, makes DEMIX highly efficient and scalable. It should also be noted that real ultrasound acquisitions are inherently noisy. Thus it is infeasible to obtain clean, noiseless ground truth data. For assessing the risk of overfitting, we evaluated the trained model on independent real ultrasound images through a downstream segmentation task. The improvements observed in metrics demonstrate that the model generalizes effectively and does not overfit to the noise present in the training ground truths.

## 7 Conclusions

This work presents DEMIX, a novel, dual-encoder denoising framework inspired by the diffusion models for mixed noise reduction along with non-uniform, spatially varying PSF-induced distortions. DEMIX demonstrates superior qualitative and quantitative performance for both datasets compared to state-of-the-art image denoising algorithms. The masked gated fusion mechanism makes the model capable of denoising additive, multiplicative, or any combination of additive or multiplicative noise components. This method holds promise for denoising a wide class of images from various applications such as microscopy, astronomy, and sonar, where images are degraded by signal-dependent and signal-independent noise components and additionally degraded by the system PSF.

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
