# OpenReview forum: "DEMIX: Dual-Encoder Latent Masking Framework for Mixed Noise Reduction in Ultrasound Imaging"
_TMLR — Rejected by TMLR_

### Review · Reviewer_EVT5 · 2026-02-04

**Summary Of Contributions:**

This paper targets on the unified denoising problem in the ultrasound imaging, which aims to restore the image from three types of noise (multiplicative noise, additive noise and PSF). The paper design a framework to solve the problem, including a noise encoder, gated design for additive and multiplicative noises and a MS-SSIM loss. These three designs are built together to get the DEMIX method that outperforms all the previous state-of-the-act baselines.

**Audience:**

Yes

**Audience Explanation:**

The paper is about ultrasound imaging restoration, which is an important application of machine learning.

**Broader Impact Concerns:**

No border impact concerns.

**Claims And Evidence:**

Yes

**Claims Explanation:**

The qualitative and quantitative experiments, together with several ablation studies sufficiently verify the model design's (noise encoder, gated fusion and the MS-SSIM loss) excellent performance in this unified noise setting.

**Requested Changes:**

1. It is better to show also demonstrate some case that deploy your method in some real-world ultrasound images to show its generalization ability.
2. It is better to show your method‘s performance on the cases that just have one or two noises, compared to the baseline methods in Table1.
3. You can show some evidence that in real world, some ultrasound images actually conclude this type of unified noise, to show that your targeted problem is practical.

---

> ### Author Response · Authors · 2026-03-09
> **Responses to the requested changes by Reviewer EVT5**
>
> 1. We agree that evaluating DEMIX on real-world ultrasound images would further strengthen the paper. In the original manuscript, Table 3 shows the performance of DEMIX on a real ultrasound dataset. We evaluated the denoising performance based on the downstream segmentation task. Columns 2 and 3 show the Jaccard Similarity (JS) and the Dice Score (Dice) on the original noisy real dataset. Columns 4 and 5 show the JS and Dice scores when the segmentation model is trained with denoised images. Since ground truth noiseless images are unavailable for the real dataset, we provided the evaluation scores on the downstream segmentation task as an evaluation measure. These results demonstrate the generalization ability of DEMIX for real-world datasets. We will highlight this in the revised manuscript.
>
> 2. We thank the reviewer for this suggestion. Although DEMIX is proposed for the mixed noise setting, we acknowledge the importance of validating its performance under simpler noise settings. In the revised manuscript, we will add the quantitative results for individual noises and compare them with the baselines in Table 1.
>
> 3. We agree that motivating the unified noise application is important. In the revised manuscript, we will clarify that real ultrasound images suffer from a combination of signal-independent sensor noise, signal-dependent speckle and spatial blur introduced by the point spread function. When the transmitted ultrasonic waves get reflected from the different tissue boundaries, the phase and the amplitude change for each wave while the frequency remains the same. When the waves from a large number of reflectors are superimposed, the resulting expression can be expressed as the product of the true underlying signal and speckle [1]. Specifically for B-mode ultrasound images, the speckled signal represents the signature of the tissue microstructural content, where the resolution is determined by the system PSF [2]. Additionally, in other coherent imaging systems, the forward model is defined as the speckled signal convolved with the intensity PSF [3]. We will add these references from the supporting literature to highlight the relevance of the proposed problem.
>
> References:
> [1] Goodman, Joseph W. Speckle phenomena in optics: theory and applications. Roberts and company Publishers, 2007.
>
> [2] Cloutier, Guy, et al. "Quantitative ultrasound imaging of soft biological tissues: a primer for radiologists and medical physicists." Insights into Imaging 12.1 (2021): 127.
>
> [3] Lee, KyeoReh, et al. "Speckle-based X-ray microtomography via preconditioned Wirtinger flow." Light: Science & Applications 15.1 (2026): 121.

---

### Review · Reviewer_u7zw · 2026-02-11

**Summary Of Contributions:**

This paper focuses on the removal of mixed noise (combination of Gaussian noise and speckle noise) that appears in the context of ultrasound imaging. The authors propose a neural network for mixed noise removal, whose construction is inspired by diffusion, which has an encoding-decoding scheme with a bottleneck in-between, with a so-called ``dual encoder'' that processes separately features from additive noise and features from multiplicative noise. The output of this dual encoder is later combined in a gated fusion block. The authors provide quantitative experiments on two datasets (ultrasound and phantom) and show that the proposed network outperforms several state-of-the-art denoising algorithms on this specific task. An ablation study is also provided to show the importance of noise encoding, gated fusion, and of the loss chosen for learning.

**Additional Comments:**

1) Some choices of notations do not help the reader's understanding. For example: $\alpha \beta$ is widely used as the multiplication of $\alpha$ and $\beta$ and not a more specific combination. The letter $\beta$ is also used on page 7 to refer to the Sigmoid function.

2) In the dual encoder, according to Fig. 2a, the same noisy image is sent to the multiplicative/additive encoder. The difference just lies in the injection on both sides of the true level of multiplicative/additive noise. However, for image denoisers, we often observed that blind denoisers can reach performance somehow close to the same non-blind version. So, how can it be expected here that, based on the same noisy image, the encoders $E_{mul}, E_{add}$ are indeed helpful to really separate the two noise components? \\
  About that: the caption mentions that these encoders take noise schedules $\alpha, \beta$ as inputs. But is it the whole schedule or only the noise level $\alpha_t, \beta_t$ for a given $t$?
3) I am not sure to understand Fig. 4a. Why is there "timesteps'' (plural) in abscissa here? Are there several timesteps considered?

**Audience:**

Yes

**Audience Explanation:**

However, the noise combination considered in this denoising problem seems quite unusual. I think that not many researchers are currently working on this type of mixed noise.

**Broader Impact Concerns:**

No concern

**Claims And Evidence:**

No

**Claims Explanation:**

1) The authors seem to focus on mixed noise removal. However, as illustrated on Fig. 1, the system integrates a PSF modeling, which is another degradation of the input signal. Therefore, the frontier with deconvolution appears difficult to grasp. To the best of my knowledge, the inverse problem corresponding to Fig. 1 is generally called deconvolution and not denoising.

2) Many notations in the paper are unclear.
  In particular, I was unable to understand the used acquisition model (with PSF and mixed noise) because the notations for the PSF and noises do not appear clear enough to me. For example:
- How can two functions $p_1(i), p_2(j)$ (with two different variables) can be combined with a convolution to get a 2-dimensional PSF?
- Why is there a convolution in Equation (3) and why the variables $\sigma_x, \sigma_y$ replace $i,j$?
- What is the meaning of $\sigma_x, \sigma_y \sim \psi$ in Algorithm 1?
- Equation (6) seems unusual: Is $I_0(1+\eta_t^m)$ a multiplication? If I'm not mistaken, this is not the standard way of modeling speckle noise, which usually rely on some multiplicative noise with Gamma distribution, or equivalently in log-domain with an additive Fischer-Tipett noise (see e.g. [3]).


3) The description of the diffusion process appears unclear to me. In diffusion papers, the probabilistic modeling of diffused density $(q_t)$ is often inspired by an interpolation with the data distribution and pure noise (so that the backward generative process can start form pure noise). Here, for denoising, the authors give no such description and invoking "diffusion'' thus seems useless. Besides, $t$ is said to be the "noise level'', but the noise levels are actually $\alpha_t, \beta_t$ whose formula is not given immediatly (but one can understand before equation (8) that they are taken to be linear). Also, the notations $q$ around equations (14), (15) are not introduced, and equation (16) seems wrong because the proposed distribution is actually $q(I_{t-1}|I_0, I_t)$ and not $q(I_{t-1}|I_t)$.

4) The connection with existing work is difficult to understand, for several reasons:
  - If the problem is to perform PSF inversion in a context of mixed noise, it could have been helpful to explicitly write the corresponding data-fidelity and to look for existing optimization methods that can solve such problems.
  - The authors say that the network is inspired by diffusion. However, diffusion models are designed for generative modeling and not directly for denoising. Actually, the loss used to train diffusion models can be seen as a weighted squared $L^2$ loss, that is, a loss similar as the one used for Gaussian denoising. However, the "diffusion point of view'' does not help to improve denoising performance.
  - Speckle noise is frequently encountered in the context of radar imaging, and many interesting contributions have been proposed to treat speckle noise in this field (see for example [1-4], among many others). It seems surprising that the authors only briefly evoked the relation to SAR imaging at the end of Section 2, and did not try to make connections with this line of work.

5) The quantative evaluation proposed in Section 5 (in particular Table 1) does not seem completely fair, because the other considered denoising techniques (except maybe SDDPM and PSF-SRDN) were not designed to handle this specific mixed noise problem. For methods based on learned neural networks (e.g. DnCNN, SwinIR, Pureformer etc), a fair comparison would at least require to retrain the other architectures on the same dataset, in the same conditions.

6) I do not fully  understand Fig. 5. The subsequent following claim from Sec. 6 does not appear well supported: "The proposed model is generalizable and can disentangle the different noise components.''

7) In Section 6, the authors said that "DEMIX does not require that the noise level or the PSF distortion be known beforehand.'' I do not understand why this is the case, because the DEMIX network takes $\alpha, \beta$ and the PSF as input. The next sentence also seems very unclear to me (``the model is trained to dynamically adapt to different levels of noise'')

References:

[1] J. Lee, “Digital image smoothing and the sigma filter,” Computer vision, graphics, and image processing, vol. 24, no. 2, pp. 255–269, 1983.

[2] G. Fracastoro, E. Magli, G. Poggi, G. Scarpa, D. Valsesia, and L. Verdoliva, “Deep learning methods for SAR image despeckling: trends and perspectives,” arXiv preprint arXiv:2012.05508, 2020.

[3] Dalsasso, E., Denis, L., and Tupin, F. ``As if by magic: Self-supervised training of deep despeckling networks with MERLIN''. IEEE Transactions on Geoscience and Remote Sensing, 60, 1-13, 2021.

[4] Zelong Wang, Jialing Han, and Chenlin Zhang. ``Diffusion posterior sampling for SAR despeckling''. IEEE Transactions on Geoscience and Remote Sensing, 2025.

**Requested Changes:**

1) The authors should give better motivation for the mixed noise model explained in Sec. 4.1 and explain how it is related to the standard modeling of speckle noise?

2) The positioning of this contribution with respect to the existing literature should be clarified. If the considered problem can indeed be cast as an inverse problem, it can be accounted for with a data-fidelity that would include the PSF and both sources of noise. There are many techniques (e.g. plug-and-play algorithms, diffusion-based or flow-matching-based solvers) that could deal with this specific inverse problem.
Otherwise, it may be understood that this mixed noise problem is too difficult for such methods and requires the design of specific networks. But in this case, the authors should provide a more comprehensive evaluation with caution retraining of all considered networks on the same dataset.

3) Some more details could be given in the paragraph ("Denoising Framework'' of page 7) about the precise network architecture. "dual encoder UNet'' is not enough to allow one to reproduce the paper's results.

4) The authors could better explain how were obtained the noisy test images. In this case, is the degradation real or simulated?

---

> ### Author Response · Authors · 2026-03-09
> **Responses to the comments by Reviewer u7zw Part 1**
>
> 1. We agree that integrating the PSF modeling into the overall system places the proposed method within the broad category of image deconvolution. In coherent imaging systems, the observed measurements are degraded by signal-dependent speckle, signal-independent sensor noise and system-specific PSF-induced distortions.
>
> In this work, we address the joint image restoration problem that accounts for both noise and PSF-induced spatial blur. We refer to this method as a denoising framework due to the dominance of speckle in ultrasound imaging. We will revise the manuscript to clarify this and explicitly state that the proposed method addresses the joint deconvolution and denoising under signal-dependent and signal-independent mixed noise degradation.
>
> 2a. $p_1(i), p_2(j)$ represent the one-dimensional PSFs along the lateral and axial directions of the ultrasonic system, respectively. The two-dimensional overall PSF is expressed as the convolution of $p_1(i)$ and $p_2(j)$. The assumption of separability is used in ultrasound imaging, as the axial and lateral responses are governed by different physical mechanisms [1]. We will clarify this in the revised manuscript.
>
> 2b. Computationally, the PSF can be implemented as a convolution filter which can be used to obtain the spatially blurred image. The PSF is defined over spatial coordinates $i, j$, and $\sigma_x$ and $\sigma_y$ control the extent of blur induced by the PSF in both directions. For a fixed PSF kernel size (such as $5 \times5$ or $3 \times3$), the individual filter matrix components vary when $\sigma_x$ and $\sigma_y$ change. The PSF remains indexed by $i, j$, but the numerical values are functions of $\sigma_x$ and $\sigma_y$. We will clarify this in the revised manuscript.
>
> 2c. As mentioned in Section 4.2, "PSF Encoder" of the original manuscript, $\psi \in \mathbf{R}^{2 \times m}$ captures the complete spectrum of PSF degradation levels and $m$ represents all degradation levels for each of $\sigma_x, \sigma_y$. During each training iteration, the degradation levels $\sigma_x, \sigma_y$ are chosen randomly for each input image. Thus, we use the notation $\sigma_x, \sigma_y \sim \psi$ in Algorithm 1. We will clarify this in the revised manuscript.
>
>
> References:
>
> [1]. Yu, Yongjian, and Scott T. Acton. "Speckle reducing anisotropic diffusion." IEEE Transactions on image processing 11.11 (2002): 1260-1270.

---

> ### Author Response · Authors · 2026-03-09
> **Responses to the comments by Reviewer u7zw Part 2**
>
> 2d. In $I_0( 1+ \eta_t^m)$, it is a multiplication. The Gaussian approximation of speckle noise proposed in the paper has been motivated by several practical and empirical considerations, as stated below.
>
> i. Distribution symmetry: For ultrasound images having a large number of scatterers in one resolution element and the phases of the scattered waves are uniformly distributed between $0$ and $2\pi$, the resultant phasor can be represented as two independent Gaussian density functions with zero mean and non-zero variance. This can be approximated as the Gaussian distribution when a strong distributed specular component is combined with a weak diffuse component of the backscatter [2].
>
> ii.  High SNR regions: Image regions that have fully developed speckle before logarithmic compression can be modeled by a Rician distribution. When the signal-to-noise ratio (SNR) is high, the Rician distribution can be approximated as a Gaussian distribution [3]. In our paper, we assume that the speckle is fully formed and the SNR is sufficiently high such that the Gaussian approximation holds.
>
> iii.  Simplicity of Gaussian models: Gaussian models have closed-form expressions, making them particularly suitable for learning-based or variational image reconstruction frameworks. In contrast, incorporating Gamma or Rician noise models would require significantly more computation, as their log-likelihoods involve complex functions such as modified Bessel or digamma functions [4]. These introduce numerical challenges and increase the overhead of backpropagation in deep neural networks [5, 6]. Moreover, these functions are generally non-convex, which makes the optimization problem more challenging. This practical consideration partly motivates our choice of Gaussian approximation, which allows for convex formulations and efficient gradient-based methods.
>
> Further, to compare the denoising performance of the Gaussian model and the Gamma model, we trained a denoiser for just the multiplicative noise corruption. The UNet has the same encoder and decoder of DEMIX, without PSF encodings. For noisy images with SNR 22.25 dB $\mapsto$  (32.81 dB, 0.895), 13.91 dB $\mapsto$ (31.71 dB, 0.883), 8.5 dB $\mapsto$ (29.95 dB, 0.854), 6.02 dB $\mapsto$ (28.72 dB, 0.828) where (PSNR, SSIM) are given for denoised images. We evaluated speckled images following Gamma noised using the same pretrained model: 13.80 dB $\mapsto$ (31.59 dB, 0.88), 9.03 dB $\mapsto$ (28.86 dB, 0.854), 6.02 dB $\mapsto$ (26.24 dB, 0.825). We can see that even though we trained the denoiser on the noise model $I_0( 1+ \eta_t^m)$, the performance on the Gamma noise model is similar.
>
> References:
>
> [2]. Wagner, Robert F., et al. "Statistics of speckle in ultrasound B-scans." IEEE Transactions on sonics and ultrasonics 30.3 (2005): 156-163.
>
> [3]. Krissian, Karl, et al. "Oriented speckle reducing anisotropic diffusion." IEEE Transactions on Image Processing 16.5 (2007): 1412-1424.
>
> [4] Sijbers, Jan, et al. "Maximum-likelihood estimation of Rician distribution parameters." IEEE Transactions on Medical Imaging 17.3 (1998): 357-361.
>
> [5] Bishop, Christopher M., and Nasser M. Nasrabadi. Pattern recognition and machine learning. Vol. 4. No. 4. New York: springer, 2006.
>
> [6] Zhang, Kai, et al. "Beyond a gaussian denoiser: Residual learning of deep cnn for image denoising." IEEE transactions on image processing 26.7 (2017): 3142-3155.

---

> ### Author Response · Authors · 2026-03-09
> **Responses to the comments by Reviewer u7zw Part 3**
>
> 3. In case of standard DDPM-based frameworks, noise is additive and $(q_t)$ is inspired by an interpolation with the data distribution and pure noise. However, in this case, the noise is signal-dependent, and the final noisy version $I_T$ remains signal-dependent. However, invoking a diffusion-based formulation provides a principled mechanism to construct a sequence of progressively degraded images governed by a well-defined forward stochastic process. The proposed forward process is derived from the statistical model of mixed noise and allows the intermediate distributions to remain physically meaningful and consistent with the underlying imaging model. The reverse process learns to approximate the corresponding inverse transformation under the correct noise statistics, which preserves the physics of the acquisition process while retaining the tractability and stability advantages of diffusion-based optimization frameworks.
>
> We agree that even though $t$ is said to be the "noise level", it is used to index $\alpha_t, \beta_t$, which represent the true levels of multiplicative and additive noise components. Thus, the effective noise level is governed by these parameters. We will clarify this distinction in the revised manuscript.
>
> We apologize for not introducing Equations 14 and 15 in the original manuscript. These represent the single-step and t-step transition processes of the forward process. We will clarify this in the revised manuscript. \\
>
> We further thank the reviewer for pointing out the correction of Equation 16, which is indeed $q(I_{t-1}|I_0, I_t)$. However, the rest of the formulation remains the same. The reverse process is modeled to follow $q(I_{t-1}|I_0, I_t)$ as closely as possible and modeled as $f_{\theta}(I_{t-1}|I_t)$.
>
> 4a. Even though the images are degraded by PSF-induced distortions, the signal-dependent multiplicative noise component dominates the distortions in the final observations. We aim to propose a unified image restoration framework that accounts for all three sources of degradation. Instead of deriving the explicit data-fidelity term and solving the resulting inverse problem with classical optimization techniques, we adopt a diffusion-inspired formulation. This helps the model learn the reverse mapping for joint deconvolution and denoising inverse problem under mixed noise without any handcrafted regularization term.
>
> 4b. We agree that diffusion models are designed for generative modeling. However, these are at their core image denoising models which eventually learn to generate data. Even during inference, these models iteratively denoise the noisy inputs and finally end up with a clean image. We have adopted the framework and replaced the additive noise with mixed noise and PSF-induced degradations.
>
> With the diffusion-inspired framework, the forward process helps corrupt the input images following the mixed noise statistics and incorporates PSF-induced distortions, thus making the model familiar with different corruption strengths. This makes the model learn to denoise images corrupted to a different extent. We will clarify this distinction in the revised manuscript to better articulate the methodological contribution.
>
> 4c. We thank the reviewer for highlighting the rich literature on speckle reduction in SAR imaging. Speckle is observed in all coherent imaging systems, including radar, electron microscopy, sonar, and laser. However, to the best of our knowledge, we are the first to propose a joint framework to reduce the corruptions caused by signal-dependent and signal-independent noise components, along with PSF-induced distortions. Additionally, DEMIX does not need any handcrafted priors or iterative processing and can restore the noisy inputs in a single forward pass through the network. This allows efficient inference while maintaining consistency with the overall degradation model. We will discuss the connections and distinctions in the revised manuscript.
>
> 5. We would like to clarify that all learning-based architectures used in the comparisons have been trained on exactly the same conditions with the same datasets. We have not used any pretrained models. Thus, each architecture was optimized specifically for the same degradation model, ensuring a fair comparison. The observed performance can be attributed to the knowledge of the noise schedules encoded in the network architectures. Additionally, evaluation scores demonstrate that UNet-based convolutional architectures tend to perform better than transformer architectures.

---

> ### Author Response · Authors · 2026-03-09
> **Responses to the comments by Reviewer u7zw Part 4**
>
> 6. The objective of Fig. 5 is to evaluate whether a model trained exclusively on mixed-noise degradations (multiplicative + additive noise + PSF distortion) can generalize to settings where only a subset of these noise components is present. The model has been trained only on mixed noise data. To evaluate generalizability, we tested the model for three different noise combinations: (i) mixed noise, (ii) multiplicative noise, and (iii) additive noise. We evaluate the qualitative and quantitative results for all three cases. This setup introduces a controlled distribution shift since the noise statistics differ from those seen during training. Despite the shift, the model achieves strong qualitative (Fig 5a) and quantitative (Fig 5b) performance.
>
> The results demonstrate that the model does not overfit to the joint distribution of the different degradation sources and can be adapted when either of the degradation components are absent. Thus, we claim that the model effectively learns to disentangle the underlying noise components.
>
> We will revise Section 6 to clarify this and explicitly connect Fig 5 to this claim.
>
> 7. Our statement refers to the fact that the model does not require the exact noise parameters or PSF corresponding to an input image. Unlike DDPMs, the proposed architecture does not require knowledge of the exact noise level. Instead, it takes as input the entire noise schedules and the complete spectrum of PSF distortions along both axial and lateral directions. The model learns to estimate the noise strength present in the input images and adapts the denoising performance to effectively reduce noise while maintaining the fine structures. The noise schedules and the PSF information embedded in the architecture comprise the continuous range of degradations, allowing the model to generalize without requiring manual tuning or exact noise parameters corresponding to the input image.
>
> Additional comments:
> 1. We will update the notation in the revised manuscript to avoid confusion.
>
> 2. Both encoders receive the same noisy image as the input, but are conditioned by different noise schedules. The network does not have access to the true level of additive/multiplicative noise and can only access the entire noise schedule. Each encoder learns to associate distinct noise characteristics in the corrupted image. The encoders are not expected to perform an exact analytical separation of the two noise components. Rather, the dual-encoder architecture encourages the network to learn feature representations that are more sensitive to signal-dependent or signal-independent noise components.
>
> In addition, the encoders are never conditioned on individual noise levels $\alpha_t, \beta_t$, and can only access all the noise schedules $\alpha, \beta$. We will clarify this in the revised manuscript.
>
> 3. We thank the reviewer for pointing this out. In Fig 4a, "timesteps" refers to the diffusion index $t \in \{1, 2, ...T\}.$ Each point on the horizontal axis corresponds to a single timestep value. We evaluate the performance of the model by denoising noisy inputs corresponding to different levels of $\alpha_t, \beta_t$ and plot the results as a function of $t$. We will clarify this in the revised manuscript.

---

> ### Author Response · Authors · 2026-03-09
> **Responses to requested changes by Reviewer u7zw**
>
> 1. In the revised manuscript, we will expand the discussion in the introduction to clarify that real ultrasound images suffer from a combination of signal-independent sensor noise, signal-dependent speckle and spatial blur introduced by the point spread function. When the transmitted ultrasonic waves get reflected from the different tissue boundaries, the phase and the amplitude changes for each wave while the frequency remains the same. When the waves from a large number of reflectors are superimposed, the resulting expression can be expressed as the product of the true underlying signal and speckle [1]. Specifically for B-mode ultrasound images, the speckled signal represents the signature of the tissue microstructural cellular content, where the resolution is determined by the system PSF [2]. Additionally, in other coherent imaging systems, the forward model is defined as speckled signal convolved with the intensity PSF [3].
>
> 2. We thank the reviewer for raising this important point. Plug and play (PnP) frameworks involve iterative strategies. The methods that have been proposed for enhancing speckle corrupted images with PnP priors also employ iterative solvers [4, 5]. In contrast, the proposed framework enhances degraded inputs in a single forward pass without requiring iterative processing.  Similarly, standard diffusion models assume additive Gaussian noise and have an iterative inference strategy. Noise is progressively removed iteratively during inference, which increases the computational complexity. Very few algorithms exist for utilizing flow matching algorithms for enhancing speckle corrupted images [6]. However, these algorithms are also iterative and require accurately specifying a probability path to map noise to target distribution.
>
> In contrast, DEMIX learns a unified restoration strategy without requiring iterative inference. We will clarify this in the revised manuscript.
>
> 3. We thank the reviewer for this suggestion. Section 4.2 of the original manuscript describes the different components of the proposed framework. We will include additional details in the revised manuscript. We will also release the complete implementation to facilitate reproducibility upon acceptance.
>
> 4. We thank the reviewer for this question. The training and validation images have simulated noise for evaluating the method on different degradation levels. For evaluating the method on real degradations, we report the downstream evaluation in Table 3 of the original manuscript. Even though the model was trained on simulated degradations, Table 3 demonstrates that the method is capable of enhancing real ultrasound images. We will clarify this in the revised manuscript.
>
> References:
>
> [1]. Goodman, Joseph W. Speckle phenomena in optics: theory and applications. Roberts and company Publishers, 2007.
>
> [2]. Cloutier, Guy, et al. "Quantitative ultrasound imaging of soft biological tissues: a primer for radiologists and medical physicists." Insights into Imaging 12.1 (2021): 127.
>
> [3]. Lee, KyeoReh, et al. "Speckle-based X-ray microtomography via preconditioned Wirtinger flow." Light: Science & Applications 15.1 (2026): 121.
>
> [4]. Baraha, Satyakam, and Ajit Kumar Sahoo. "Plug-and-play priors enabled SAR image inpainting in the presence of speckle noise." 2020 IEEE 17th India Council International Conference (INDICON). IEEE, 2020.
>
> [5]. Baraha, Satyakam, and Ajit Kumar Sahoo. "Speckle removal using dictionary learning and PnP-based fast iterative shrinkage threshold algorithm." IEEE Geoscience and Remote Sensing Letters 20 (2023): 1-5.
>
> [6]. van de Schaft, Vincent, and Ruud JG van Sloun. "Ultrasound speckle suppression and denoising using MRI-derived normalizing flow priors." arXiv preprint arXiv:2112.13110 (2021).

---

### Review · Reviewer_3CSJ · 2026-03-02

**Summary Of Contributions:**

The paper presents a denoising architecture and training procedure for ultrasound images. The corruption process is modeled as multiplicative and additive noise, and a convolution with an anisotropic point spread function (PSF). The denoiser network is trained on a set of noise levels and PSF sizes, and its architecture comprises two encoders (one for multiplicative noise and the other for additive noise), one embedding block that conditions on the multiplicative and additive noise levels, and one embedding layer for the PSF sizes.

**Strenghts**
- tackles the problem of multiplicative noise on ultrasound images, which is often overlooked by most denoising methods.

**Weaknesses**
- The presentation should be improved to make the paper clearer and more accessible:
    - The forward process and reverse process are very unclear to me. What is the goal of this section? It is unclear to me whether the model is simply trained at different noise levels and blur parameters (in which case the links to diffusion models are very mild and could be omitted for clarity) or if the model is actually evaluated using the reverse process in eqs. (17) and (18)? If the latter is the case, I don't understand why the diffusion model is needed if only distortion metrics are evaluated (PSNR, SSIM), and no perceptual metrics are considered (e.g., FID scores) - see *The perception-distortion tradeoff* by Blau & Michaeli. If only distortion metrics are needed, the use of a diffusion model should be better justified.
    - Some paragraphs need rewriting or could be removed. For example, the "Efficient training" section revolves around using patches for training instead of full images, which is used in most denoising methods in the literature. Many statements are not backed by experiments nor references - for example "Instead of learning the distribution of images, DEMIX learns the local noise characteristics and reverses the degradation process.": There is no proof of this in the paper - I would argue that denoising patches requires learning the distribution of patches. A good denoiser requires some modelling of the target signal distribution, and is not solely a function of the noise distribution.
    - The mathematical notation is highly inconsistent:
        - Bold is used in random places (e.g., for the mean and covariance of the normal distribution between equations 6 and 7, but there is no bold in the normal distribution in Algorithm 1). I would recommend using bold for vectors and matrices.
        - Equations 1 and 2 are unclear - what is $i$ and $j$? Pixels? What are their ranges?. Why is (3) replacing $i$ and $j$ by $\sigma$ values? The construction of the PSF is quite confusing to me.
        - In Algorithm 1, $\sigma_x$ and $\sigma_y$ are sampled from $\psi$, but then $psi$ is given to the model... How the distribution $psi$ is given to the model?
        - (a detail) Please use backslash | for norms (not double ||)

- It is unclear how the different baselines were evaluated. Did the authors apply the same training (Algorithm 1) to all models? Some models (e.g., DRUNet) can receive conditioning on the noise parameters - was this included when the authors retrained these methods? Moreover, the step size, batch size, etc., can have a strong impact on the models. What values did the authors use for each model? Were these values ablated? It would be good to understand if the performance gap arises from a better architecture or just the fact that the proposed model is conditioned on the noise and PSF parameters, whereas the other competing methods are not.

- It would be good to add an ablation where one of the encoders is removed - the choice of having two different encoders is a bit non-standard, and it is unclear whether this is actually necessary.

- The ultrasound application does not seem to be realistic enough:
     - The paper considers a PSF applied after adding multiplicative noise, whereas most ultrasound papers consider blurring before the multiplicative noise.
     - It is unclear which datasets were used. The first reference (LLC, 1999) is "MultiMedia LLC. MS Windows NT kernel description, 1999. URL http://splab.cz/en/download/ databaze/ultrasound." and the link doesn't even work. The second reference (Zhang & Zhang, 2022) does not contribute a dataset, but uses other existing datasets. Also, it doesn't mention common carotid arteries (as suggested in the papers). These issues lead me to think that some of the content in the paper might be LLM-generated.
   - The paper evaluates its performance on synthetically added noise with known noise paramers - since the paper is focused on the application of ultrasound, I would expect some evaluations on real images with real noise.
   - The method trains the model using ultrasound images as ground-truths, which seem to have significant amounts of noise. There is a risk of overfitting the real noise - a discussion and evaluation around this point is lacking.
   - There is no discussion of the specific ultrasound modality that the authors want to tackle - I imagine this is B-mode images, but a brief description of different types of ultrasound modalities would add value to the paper.

**Audience:**

No

**Audience Explanation:**

Not in the current form - I think the paper requires a major rewrite and clarification on the diffusion part and how baselines and datasets are considered.

**Broader Impact Concerns:**

No specific concerns.

**Claims And Evidence:**

No

**Claims Explanation:**

The claims in the submission are not sufficiently supported by accurate, convincing, or clear evidence. Several key issues undermine the credibility of the presented results and methodology:

1. The forward and reverse processes are ambiguously described, making it difficult to assess whether the model is trained at various noise levels and blur parameters or evaluated using the reverse process (Eqs. 17–18). The justification for employing a diffusion model is weak, particularly since only distortion metrics (PSNR, SSIM) are reported. The absence of perceptual metrics (e.g., FID scores) further weakens the claim that the model effectively addresses perceptual quality.

2. Many statements lack experimental validation or references. For example, the assertion that "DEMIX learns the local noise characteristics and reverses the degradation process" is not empirically supported. Denoising patches inherently involves learning the distribution of patches, not just noise characteristics. Without evidence, such claims remain speculative.

3. The notation is inconsistent and confusing:
   - Bold formatting is applied arbitrarily (e.g., mean and covariance in Eqs. 6–7 but not in Algorithm 1).
   - Eqs. 1–2 are unclear regarding the meaning of *i* and *j* (pixel indices?) and their ranges.
   - The construction of the PSF is not clearly explained, and the role of ψ in Algorithm 1 is ambiguous.

4. The evaluation of baselines is not transparent. It is unclear whether all models were trained using Algorithm 1 or if noise parameter conditioning was applied to baselines like DRUNet. Training parameters (e.g., step size, batch size) are not specified, making it difficult to determine if performance gains stem from architectural improvements or simply from conditioning on noise and PSF parameters.

5. The necessity of using two separate encoders is not justified. An ablation study removing one encoder would clarify whether this design choice is essential or redundant.

6. The ultrasound application lacks realism:
   - The PSF is applied after multiplicative noise, which contradicts most ultrasound models where blurring precedes multiplicative noise.
   - The datasets used are unclear, with broken references and no mention of common carotid arteries. This raises concerns about the validity of the data.
   - Performance is evaluated on synthetic noise with known parameters, rather than real ultrasound images with actual noise.
   - Training on noisy ultrasound images as ground truth risks overfitting to real noise, yet this issue is not discussed or evaluated.

7. The paper does not specify the ultrasound modality addressed (e.g., B-mode). A brief discussion of different ultrasound modalities would provide context and enhance the paper’s relevance.

In summary, the submission’s claims are not adequately supported by clear, accurate, or convincing evidence. The methodology lacks transparency, the mathematical presentation is inconsistent, and the evaluation is insufficiently rigorous for the claims made. Addressing these issues would significantly improve the credibility of the work.

**Requested Changes:**

1. **Clarify methodology**
   - Clearly distinguish between training and evaluation processes (forward/reverse).
   - Justify the use of a diffusion model, especially if only distortion metrics (PSNR, SSIM) are reported. Include perceptual metrics (e.g., FID) if perceptual quality is claimed.

2. **Revise unsubstantiated claims**
   - Remove or support statements lacking evidence (e.g., "DEMIX learns local noise characteristics").
   - Clarify the role of patch-based training and its relationship to signal/noise distribution modeling.

3. **Fix mathematical notation**
   - Standardize bold usage (vectors/matrices only).
   - Define *i*, *j*, and σ ranges in Eqs. 1–3.
   - Explain PSF construction and the role of ψ in Algorithm 1.
   - Use backslash \| for norms.

4. **Detail baseline evaluation**
   - Specify training protocols for all baselines (e.g., Algorithm 1, noise conditioning).
   - Report hyperparameters (step size, batch size) and ablate their impact.

5. **Add ablation studies**
   - Evaluate the necessity of dual encoders by removing one.

6. **Improve Ultrasound Application Realism**
   - Apply PSF before multiplicative noise (consistent with ultrasound literature).
   - Use real ultrasound datasets; replace broken references.
   - Evaluate on real-noise images, not just synthetic data.
   - Discuss risks of overfitting to noisy ground truth.

7. **Specify Ultrasound Modality**
   - Clarify the target modality (e.g., B-mode) and briefly contextualize other modalities.

---

> ### Author Response · Authors · 2026-03-09
> **Responses to comments by Reviewer 3CSJ Part 1**
>
> 1a. In this work, we propose a diffusion-inspired denoising framework to enhance ultrasound images degraded with mixed noise along with system-specific blur. The forward process provides a principled mechanism for generating progressively degraded observations by incorporating different levels of signal-dependent speckle noise, signal-independent noise, and PSF-induced distortions. This formulation allows the model to be trained across a range of degradation levels within a unified framework.
>
> The reverse process equations are used to formulate the training objective, where the network learns to directly predict the denoised images from degraded input. DEMIX denoises images in a single inference step, without any iterative sampling seen in generative diffusion models. Since the proposed framework is particularly aimed at enhancing the images, this method gives us an efficient inference mechanism compared to the standard formulations of diffusion models. Thus, the diffusion formulation serves as a structured way to model the degradation process and guide the training objective.
>
> 1b. We thank the reviewer for this comment. Since DEMIX is proposed for image enhancement and not for generation purposes, we focus on distortion-based evaluation metrics such as PSNR and SSIM that measure the quality of the restored images compared to the true underlying signal.  We use and respect the FID score, but in this case, we are not comparing a generated image to a real image. So, the distributional distance may not be an effective judge of enhancement, which is the application we are attempting.
>
> 2. Since DEMIX is primarily an image enhancement framework, we emphasized that the proposed framework does not need to learn the distribution of images, as required by generative models. We agree that effective denoising inevitably requires learning aspects of the underlying signal distribution in addition to the noise statistics. We will revise the wording to avoid suggesting that the model only learns the noise distribution.  We will also make the presentation more concise.
>
> 3a. We will revise the notations in the revised manuscript.
>
> 3b. In Eqs 1 and 2, $i$ and $j$ represent the rows and columns of the PSF matrix. Their ranges depend on the dimensions of the PSF matrix. Computationally, the PSF is analogous to a blur kernel and $i$ and $j$ represent the row and column indices of the kernel. $p_1(i), p_2(j)$ represent the one-dimensional PSFs along the lateral and axial directions of the ultrasonic system, respectively. The overall two-dimensional PSF is expressed as the convolution of $p_1(i)$ and $p_2(j)$. The assumption of separability is used in ultrasound imaging, as the axial and lateral responses are governed by different physical mechanisms. Given ultrasound B-mode processing, the waves are transmitted and received in the \textit{i} direction. Horizontal crosstalk between elements leads to a different PSF response in \textit{j}.
>
> The PSF is defined over spatial coordinates $i, j$, and $\sigma_x$ and $\sigma_y$ control the extent of blur induced by the PSF in both directions. For a fixed PSF kernel size (such as $5 \times5$ or $3 \times3$), the individual filter matrix components vary when $\sigma_x$ and $\sigma_y$ change. The PSF remains indexed by $i, j$, but the numerical values are functions of $\sigma_x$ and $\sigma_y$.
>
> 3c. As described in "PSF Encoder" subsection of Section 4.2 in the original manuscript, the complete spectrum of $\sigma_x$ and $\sigma_y$ are stacked together to form a matrix $\in \mathbf{R}^{2 \times m}$ to form $\psi$, where $m$ is the total number of distortion levels considered for each of $\sigma_x$ and $\sigma_y$. Thus, during each training iteration, $\sigma_x$ and $\sigma_y$ are randomly sampled for each image to select a random distortion level. To make the model dynamically assess the distortion level for each noisy input, the entire $\psi$ matrix is embedded into the different layers of both encoders.
>
> 3d. We will update the notation in the revised manuscript.
>
> 4. All models were trained under the same conditions (Section 5.1 of the original manuscript) in order to have a fair comparison. Specifically, we followed the original training guidelines for each baseline architecture while keeping the training protocol consistent across methods. The step size, batch size, and other optimization hyperparameters were kept identical for all models so that each method was optimized under the same degradation model described in Algorithm 1.
>
> We would like to clarify that DRUNet was not used to compare the performance of the proposed model. Our goal was to evaluate whether the proposed architecture can effectively learn the restoration task under the same training conditions as the competing methods. The observed performance differences arise from the modeling approach rather than differences in training configuration.

---

> ### Author Response · Authors · 2026-03-09
> **Responses to comments by Reviewer 3CSJ Part 2**
>
> 5. We used an ablated model with single encoder and the noise schedules $\alpha$ and $\beta$ were combined with the noise encoder and fed into different encoder layers. No masking was therefore needed. We report the scores for the two datasets below ($\sigma_x = 2, \sigma_y = 1.5$).
>
> Ultrasound: Dual-encoder (PSNR, SSIM) $\mapsto$ Single-encoder (PSNR, SSIM)
>
> (26.85 , 0.745) $\mapsto$ (25.43, 0.708); (24.55, 0.671) $\mapsto$ (22.94 , 0.624); (23.22, 0.629) $\mapsto$ (21.78,  0.567); (22.27, 0.586) $\mapsto$ (20.91,   0.528)
>
> Phantom: Dual-encoder (PSNR, SSIM) $\mapsto$ Single-encoder (PSNR, SSIM)
>
> (25.10, 0.664) $\mapsto$ (22.52, 0.617); (22.33, 0.607) $\mapsto$ (21.19, 0.586 ); (21.29, 0.589) $\mapsto$ (20.74, 0.577); (20.76, 0.581) $\mapsto$ (20.13, 0.571)
>
> 6a. When the transmitted ultrasonic waves get reflected from the different tissue boundaries, the phase and the amplitude changes for each wave while the frequency remains the same. When the waves from a large number of reflectors are superimposed, the resulting expression can be expressed as the product of the true underlying signal and speckle [1]. Specifically for B-mode ultrasound images, the speckled signal represents the signature of the tissue microstructural cellular content, where the resolution is determined by the system PSF [2]. Additionally, in other coherent imaging systems, the forward model is defined as speckled signal convolved with the intensity PSF [3].
>
> 6b. We sincerely apologize for the mistake. We evaluated the proposed method for ultrasound images of the common carotid artery (CCA) [4] and for those of the brachial plexus, fetal heads, and lymph nodes [5]. The latter dataset was originally part of a Kaggle competition, which cited Zhang \& Zhang (2022). However, the competition page is no longer available. We have now included the original Kaggle link for reference. The CCA dataset was previously hosted on a server that is no longer accessible. We now refer to the original data set. The images had been downloaded prior to their removal and were used directly in our experiments.
>
> We will update the manuscript to correctly cite the datasets used and update the links.
>
> 6c. Although noise was added synthetically during the training and evaluation phases, in order to evaluate the model for different levels of distortion. However, in standard DDPMs, the model does not require knowledge of the exact noise level. Instead, it takes as input the entire noise schedules and the complete spectrum of PSF distortions along both axial and lateral directions. The model learns to estimate the noise strength present in the input images and adapts the denoising performance to effectively reduce noise while maintaining the fine structures. The noise schedules and the PSF information embedded in the architecture comprise the continuous range of degradations, allowing the model to generalize without requiring manual tuning or exact noise parameters corresponding to the input image.
>
> Additionally, Table 3 shows the downstream segmentation performance in real ultrasound images with real noise. The real images were denoised, and the segmentation performance is reported for both the noisy original images and the denoised images. The improvement in Jaccard Similarity and Dice Scores demonstrates that the model is generalized to real-world data.
>
> 6d. We agree that the model was trained and evaluated on ultrasound ground-truth images that inherently contain some noise. Since real ultrasound images always include noise, it is not possible to eliminate this entirely. To assess the risk of overfitting, we evaluated the model on independent real ultrasound images through a downstream task, such as segmentation. The improvements observed in metrics demonstrate that the model generalizes effectively and does not overfit to the noise present in the training ground truths. We will include this discussion in the revised manuscript to clarify the robustness to real-world noise.
>
> 7. The proposed method was developed and evaluated specifically on B-mode ultrasound images. We will include a discussion of the target modality and other modalities.
>
> References:
>
> [1]. Goodman, Joseph W. Speckle phenomena in optics: theory and applications. Roberts and company Publishers, 2007.
>
> [2]. Cloutier, Guy, et al. "Quantitative ultrasound imaging of soft biological tissues: a primer for radiologists and medical physicists." Insights into Imaging 12.1 (2021): 127.
>
> [3]. Lee, KyeoReh, et al. "Speckle-based X-ray microtomography via preconditioned Wirtinger flow." Light: Science & Applications 15.1 (2026): 121.
>
> [4]. https://data.mendeley.com/datasets/d4xt63mgjm/1
>
> [5]. https://www.kaggle.com/competitions/ultrasound-nerve-segmentation

---

### Decision · Action_Editor_71E1 · 2026-04-16

**Recommendation:** Reject

**Additional Comments:**

I hope the authors find the reviews constructive and consider submitting a substantially revised version in the future.

**Audience:**

Yes

**Audience Explanation:**

The paper presents a denoising architecture and training procedure for ultrasound images. It models the corruption process as a combination of multiplicative noise, additive noise, and convolution with an anisotropic point spread function (PSF). The proposed denoiser is trained across a range of noise levels and PSF sizes. Architecturally, it uses two encoders to separately process multiplicative and additive noise, along with conditioning modules that embed the corresponding noise levels and PSF parameters. Experiments on ultrasound and phantom datasets show improved performance over several denoising baselines, and ablations evaluate the roles of the noise modeling, fusion design, and training loss.

**Claims And Evidence:**

No

**Claims Explanation:**

After considering the reviews and rebuttal, I am unable to recommend acceptance in its current form. The main reason is that substantial concerns remain about the clarity and technical presentation of the method.

* First, multiple reviewers remained unconvinced by the presentation of the method as a diffusion model. In particular, the role of the “reverse process” and its relationship to inference remained insufficiently clear. As noted in the reviews, if the network is applied only once at inference, the current diffusion-based framing may be more confusing than illuminating, and the paper would benefit from a clearer, more direct presentation of the method.

* Second, the experimental comparisons against baseline methods were viewed as insufficiently transparent. The reviews raised concerns that the training setup for competing methods is not clearly specified and may not support a fair apples-to-apples comparison. Since these comparisons are central to the empirical claims of the paper, this issue needs to be resolved much more carefully.

* Third, concerns about mathematical notation and exposition remain significant. Reviewers noted that several definitions, equations, and expectations are still imprecise or difficult to interpret, and that the rebuttal did not fully resolve these issues in the manuscript itself. At this stage, the presentation is not yet at the level of clarity required for acceptance.

Overall, while one reviewer was positive about the novelty and experimental scope of the work, the paper would benefit from substantial revision before reconsideration. A future version would be strengthened by addressing those three major issues.

**Resubmission Of Major Revision:**

The authors may consider submitting a major revision at a later time.